# CONSTANTLY IMPROVING IMAGE MODELS NEED CONSTANTLY IMPROVING BENCHMARKS

**Jiaxin Ge**[1]* **Grace Luo**[1]*

**Heekyung Lee**[1] **Nishant Malpani**[1] **Long Lian**[1] **XuDong Wang**[1]

**Aleksander Holynski**[1] **Trevor Darrell**[1] **Sewon Min**[1] **David M. Chan**[1]

[1]UC Berkeley

## ABSTRACT

Recent advances in image generation, often driven by proprietary systems like GPT-4o Image Gen, regularly introduce new capabilities that reshape how users interact with these models. Existing benchmarks often lag behind and fail to capture these emerging use cases, leaving a gap between community perceptions of progress and formal evaluation. To address this, we present ECHO, a framework for constructing benchmarks directly from real-world evidence of model use: social media posts that showcase novel prompts and qualitative user judgments. Applying this framework to GPT-4o Image Gen, we construct a dataset of over 31,000 prompts curated from such posts. Our analysis shows that ECHO (1) discovers creative and complex tasks absent from existing benchmarks, such as re-rendering product labels across languages or generating receipts with specified totals, (2) more clearly distinguishes state-of-the-art models from alternatives, and (3) surfaces community feedback that we use to inform the design of metrics for model quality (e.g., measuring observed shifts in color, identity, and structure). Our website is at https://echo-bench.github.io.

## 1 INTRODUCTION

When new generative image models are released, users often find new and unanticipated capabilities not captured by existing benchmarks. These capabilities are discussed on social media, where users document their interactions with new models and qualitatively discuss their performance. The release of GPT-4o Image Gen (OpenAI, 2025a) exemplified this behavior with the introduction of "Ghiblification," the style-transfer task of turning a natural image into a cartoon version emulating a particular animated studio. This new "task" was not only shared widely on social media, but used as a personal measure of model quality by many members the online community. As of today, explicit benchmarks have now been developed for this task (Jiang et al., 2025), but the benchmarks that we traditionally use to evaluate models do not have the capability to evolve *with* community feedback, and instead, must *react* to changes in a delayed cycle.

Indeed, despite significant changes in what constitutes a "good" image generation model, current popular crowdsourced text-to-image benchmarks (Wang et al., 2022; Kirstain et al., 2023) are often still tailored towards older models such as Stable Diffusion (Rombach et al., 2022), with extensive art-centric keyword lists that are not representative of now-feasible use cases. Popular image editing benchmarks (Zhang et al., 2023a;b; Liu et al., 2025) contain overly simple instructions. These instructions were challenging at their inception but do not actually require complex language understanding or reasoning. Furthermore, these tasks can already be solved by many models, new and old. This slow adaptation rate is reflected in model benchmark scores. As we see in Figure 1b, human ratings indicate that 4o Image Gen is substantially better than the current best open-source unified model (Deng et al., 2025), yet even when benchmarking on a recent image editing benchmark (Liu et al., 2025), the gap appears less significant.

With the rapid releases of new image generation models, each revealing a range of new capabilities to be tested, it has become clear that we need more responsive mechanisms for adapting benchmarks to emergent user observations. In this work we present ECHO: Extracting Community Hatched Observations, a *re-usable framework* that converts community discussion on social media into a structured benchmark. Our proposed

---

*Equal contribution. Correspondence to gejiaxin@berkeley.edu, graceluo@berkeley.edu

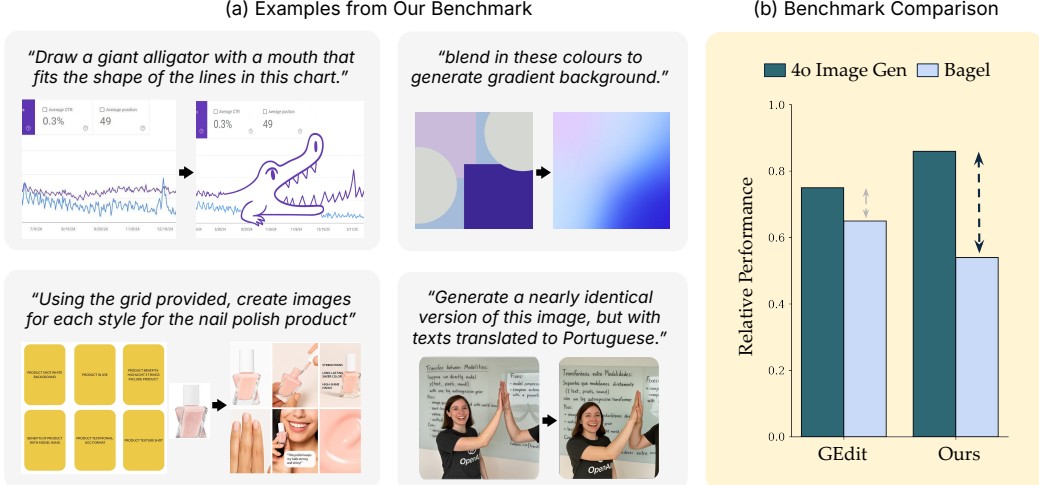

Figure 1: **ECHO** distills collective discussion about a new generative model into a structured benchmark. As a case study, we apply ECHO to GPT-4o Image Gen (OpenAI, 2025a) on Twitter/X. Left: ECHO *automatically* surfaces highly diverse and novel tasks not covered in prior benchmarks. Right: Consequently, our image-to-image split shows a 3.2x larger relative performance gap compared to a prior image editing benchmark, GEdit (Liu et al., 2025).

method bypasses the traditional "observation to benchmark" cycle, and provides us with a framework for *automatically* converting real-world ideas and capabilities surfaced by users on social media directly to metrics that we can use to measure and improve SOTA models. The ECHO framework operates by searching social media for mentions of a target model and automatically filtering for coherent image generation prompts specified via text and/or images, while extracting community insights and feedback on particular prompt capabilities. It is designed to address a number of common challenges associated with social media, including the tradeoff between post volume and relevance, the splitting of context across posts, and noisy formatting.

Using ECHO, we are able to surface and formalize a number of qualitative observations related to the most recent image generation methods. By running our framework on the 4o Image Gen release, we introduce a new dataset containing more than 31,000 user-sourced prompts which: (1) surfaces creative and complex tasks absent from existing benchmarks, (2) is more diverse and more closely resembles natural user language (contains 2.3x more unique first bigrams and are 1.2x lower in LLM perplexity), (3) better separates state-of-the-art models from prior models, and (4) *automatically* surfaces several new quantifiable indicators for image generation quality, including identity preservation and color shift, which we show can be operationalized into secondary evaluation metrics that could inform future model losses and development.

## 2 BACKGROUND & RELATED WORK

Model benchmarks often mirror the capabilities of the models themselves, and are designed by model developers in order to evaluate and understand how these models perform on tasks of interest. For example, traditional text-to-image benchmarks (Huang et al., 2023; Ghosh et al., 2023; Lee et al., 2023) and image-to-image benchmarks (Brooks et al., 2023; Wang et al., 2023; Sheynin et al., 2024; Hui et al., 2024; Zhang et al., 2023a) are not collected in-the-wild. These benchmarks contain short and overly simple instructions such as "A cat in front of a chair" or "Add fireworks in the sky" that fail to reflect real user intent, but provide strong diagnostic signal for understanding simple generative understanding.

On the other hand, community-driven benchmarks are often designed to collect real user prompts, and more closely mirror what a downstream user might desire from a model. For example, previous methods (Wang et al., 2022; Xu et al., 2023; Kirstain et al., 2023) have collected real user prompts of Stable Diffusion models from an explicit interface (Rombach et al., 2022). These benchmarks require a significant amount of human intervention to decide which user prompts to model, and how to model them. In addition, the model interface itself can lead to two further limitations: (i) prompt intent is bounded by the capabilities of the model itself, and (ii) prompt style is tailored towards the model rather than natural user language. For example, it has already been demonstrated that users will adjust their prompting behavior to account

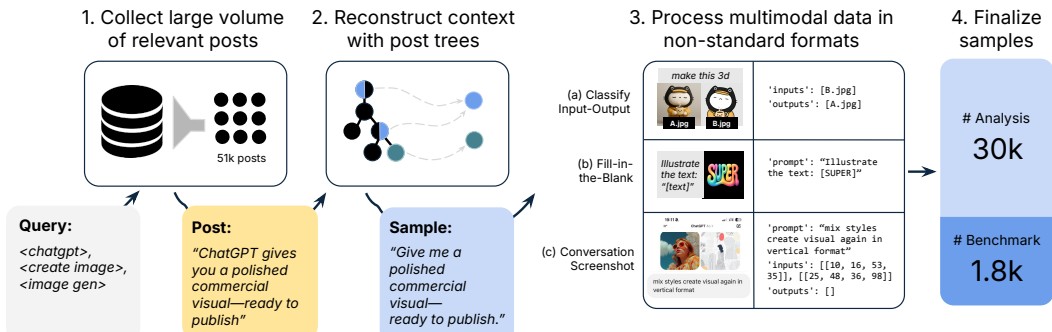

Figure 2: **ECHO Framework.** ECHO is motivated by several challenges inherent to social media. (1) We start with broad queries followed by relevance filtering, since basic querying presents a volume-relevance tradeoff. (2) We then extract prompts from these posts, making sure to utilize the full post tree, as context can be spread across posts. (3) We then apply multimodal processing, since useful data also exists in non-standard formats. (4) Finally, we reserve the highest quality data for benchmarking, while the rest is used for analysis.

for limitations of the CLIP (Radford et al., 2021) text encoder, which behaves more like a bag-of-words representation, where they use extensive sets of "phrases rather than complete sentences" (ComfyUI Wiki, 2025), with prompts like *"colorful stars,galaxies,space,artstation."* Unlike interface-collected datasets, our framework draws on prompts crafted for human audiences on social media, where the goal is to showcase creativity rather than to optimize around model quirks. While prompts inevitably reflect the capabilities of the current best models, our framework is re-runnable and can adapt as models and user behaviors evolve, reducing the risk of per-model biases.

GEdit (Liu et al., 2025) proposed scraping the internet for real image editing prompts. However, these prompts are limited by the imagination of the authors, leading to a restricted set of 11 specific single image editing tasks, such as changing the color or changing the background. Most closely related to our work, IntelligentBench (Deng et al., 2025) and KontextBench (Batifol et al., 2025) were designed to highlight the capabilities of new models released by the same authors. However, details about their data source and creation method are largely unknown, and neither benchmark is publicly available.

Outside of image generation, Chatbot Arena (Chiang et al., 2024) uses an online platform to collect use cases in the wild, incentivizing users to provide data by providing a free platform for interacting with the model. While such a process does collect real user prompts, unlike this approach, we investigate social media, which represents a notably different prompt distribution: since users are seeking reciprocal engagement, they are more incentivized to produce novel and creative examples, rather than tasks that are already well-within model capabilities.

## 3    CROWDSOURCING A BENCHMARK

Our primary goal is to distill collective discussion about a new generative model into a structured dataset. Such discussion often involves users sharing interesting prompts and outputs, novel task ideas, or commentary on model behavior. We aim to capture all of these cases, in a standardized format:

```
<input text, input image(s)*, output image, community feedback*>
```

where * denotes optional fields; the full set of data we collect is given in Table E.1. However, this objective poses several challenges:

- Collection: A large volume of relevant data is desired, which requires identifying the right platform and gathering the data.

- Processing: A non-trivial amount of processing is required, e.g., the input prompt and images may be embedded in a single screenshot or the prompt may not be written explicitly.

- Filtering: Data quality varies widely, e.g., a user may provide more general commentary or exactly document their input prompt.

We propose a framework, ECHO, that addresses these challenges, illustrated in Figure 2. Our framework first collects relevant posts (Section 3.1), converts posts into self-contained samples (Section 3.2), and finally expands coverage via multimodal processing (Section 3.3).

## 3.1 Identifying Relevant Posts

There is an inherent tradeoff between the volume of posts and their relevance. When querying with broader keywords, the average post relevance goes down, and with narrower ones, the available post pool is quickly exhausted. To address this, we implement a two-stage pipeline where we first query for a large volume of posts then use an LLM to filter irrelevant ones.

**Designing Keywords.** First, our goal is to maximize the post pool. However, we found two issues: (1) LLM-based filtering is expensive, so the pool cannot be too large, and (2) there is a temporal shift in which keywords lead to relevant posts (e.g., in the initial two weeks of the 4o Image Gen release, generic terms like "openai" often retrieve relevant posts, but later on relevancy decreases). Therefore, we use two sets of keywords to query posts within vs. outside the first two weeks of the 4o Image Gen release (see Figure E.1).

**Classifying Relevance.** We then use an LLM to classify the post text on a 5-point relevance scale (see Figure E.2). We initially collected 68k posts in total, of which 47% passed our relevance filter as 'very likely relevant' or "certainly relevant." Nearly half of collected posts pass this filter, amounting to 32k posts, indicating that our query design is fairly efficient and has a high yield rate.

## 3.2 Reconstructing Context Across Posts

Posts can be context dependent. For example, a user may write *"prompt below"* in the first post then include the actual prompt text in a reply. We want self-contained *samples*, characterized as: a unique prompt some user tried, community feedback towards that prompt and its resulting outputs, and a label for its quality. To achieve this, our framework attempts to collect as much of the reply tree as possible, then use this full context when processing posts into samples.

**Constructing Reply Trees.** For each post obtained via keyword query, we extract the full reply tree, or URLs pointing to the parent post or child replies. We then recursively expand the dataset by querying these discovered posts and traversing their respective reply chains, introducing 19k new posts from the replies. This procedure enables our framework to discover relevant posts that may not otherwise appear with keyword-based queries. After reply collection, each post contains ancestor chain $\mathcal{P}_\uparrow = \langle P_0,...,P_n \rangle$ and direct replies $\mathcal{C}_\downarrow = \{C_0,...,C_m\}$. We then search for the unique reply trees across all collected posts. We iterate through each post, referred to as the "main post" $P_{\text{main}}$. For every $P_i \in \mathcal{P}_\uparrow$, we attach $P_{i+1}$ as its sole child, producing the path $P_0 \to ... \to P_n \to P_{\text{main}}$. Each $C_j \in \mathcal{C}_\downarrow$ becomes a child of the main post, giving edges $P_{\text{main}} \to C_j$. Since the same posts can appear in multiple trees, we remove duplicates via URL and recursively union their children.

**Extracting Self-contained Samples.** We then use an LLM to convert trees into samples, as illustrated in Figure E.3. This processing step first identifies the spans of text corresponding to the prompt and discards any unrelated remarks, and performs minor fixups such as combining disjoint spans. Next, this step collects any commentary either from the original author or other user replies (e.g., *"amazing result,"* *"didn't work,"* etc.) as a list of community feedback. Finally, the sample is assigned one of three quality labels: "Benchmark" (high quality prompts that are coherent and show clear user intent), "Analysis" (moderate quality partial prompts or commentary that could not be associated with another sample), or "Trash" (off-topic and malformed content that should be discarded). While we want only the highest quality data for benchmarking, we are also interested in retaining any other relevant data for large-scale analysis.

## 3.3 Multimodal Processing of Posts with Images

While Section 3.2 can extract text prompts and community feedback, other metadata requires multimodal processing. This step marks the input and output images associated with each sample, updates prompts with fill-in-the-blank, and produces new samples by parsing screenshots, addressing three common cases:

**Classifying Input vs. Output Images.** There does not exist a standardized format for marking input and output images. For example, the output image could be the first or the last in a series of images, or

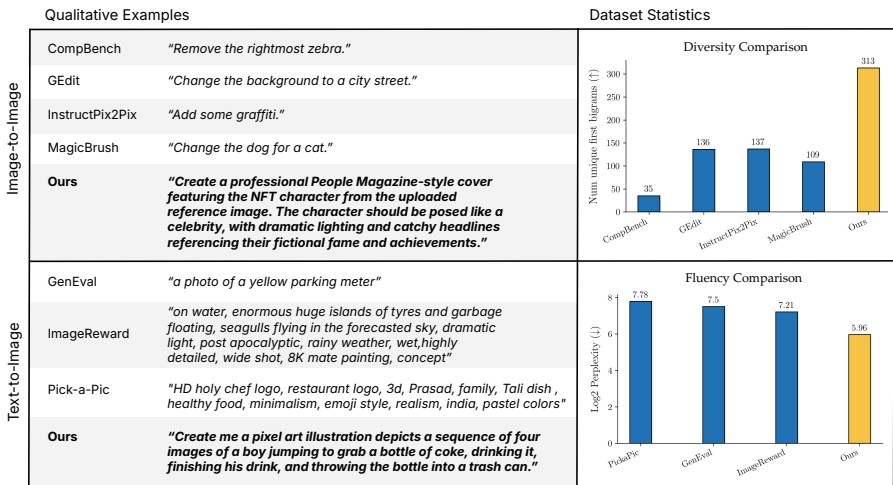

Figure 3: **Dataset Comparison.** The prompts in ECHO are significantly different from prior benchmarks. Top: the image-to-image split is more diverse and complex, with more unique first bigrams. Bottom: the text-to-image split is more fluent, as measured by perplexity under Pythia 12B (Biderman et al., 2023).

there may be irrelevant images that are neither inputs nor outputs. Nevertheless, users expect viewers to infer this distinction, and thus we use a VLM to make this same inference (see Figure E.4).

**Completing Fill-in-the-Blank.** A common user behavior is "fill-in-the-blank" prompts, where users post a template intended for commenters to infill in the replies. Keeping these templates as-is presents a problem, because they are not fully specified and effectively omit the completions that commenters find most interesting. Instead, we use a VLM to reverse-engineer these completions conditioned on the template and the images provided by commenters (see Figure E.4).

**Extracting Conversation Screenshots.** Another behavior is sharing screenshots of interactions with 4o Image Gen, which may contain prompt text, reference images, and image outputs all within the same frame. This is an especially high-quality source of data, since the inputs and outputs are exactly documented without paraphrasing or summary. Extracting the raw data requires a multi-task computer vision system that can localize images to bounding boxes, classify the sub-images as inputs vs. outputs, and detect what is prompt text vs. unrelated conversation. While one could chain together specialized models for each subtask, we instead opt for a more generalizable solution using a VLM. We first detect these cases with the general multimodal processing prompt, which is then routed to the parsing prompt depicted in Figure E.5. The VLM can not only parse the 4o Image Gen interface but also other non-standard layouts, for example side-by-side collages of input and output images. For the VLM we opt to use Qwen-2.5-VL (Bai et al., 2025), which is specifically trained for bounding box detection.

## 4    ECHO: A SOCIAL-MEDIA POST-RELEASE BENCHMARK

We initially run our framework to explore the 4o Image Gen release on Twitter/X. After the LLM quality filter in Section 3.2, we find that 20% of samples are marked as high-quality (usable for benchmarking) and 66% are marked as moderate-quality (usable for analysis). We also apply safety filtering (see more details in Section 6). This yields 30k total samples for analysis. For our final benchmark, we limit each split to up to a thousand samples, to keep the downstream costs of benchmarking (generating outputs and rating them) manageable. We then randomly sample candidates and manually inspect them, flagging low-quality examples for removal or updating their prompts, to ensure that the benchmark is the highest possible quality. This results in an image-to-image split with 777 prompt-image pairs and text-to-image split with 1000 prompts.

**ECHO Surfaces Diverse and Novel Tasks.** While most benchmarks are limited to templated image editing tasks, such as changing the background, changing the color, adding or replacing an object, as shown in Figure 3 (top left), ECHO incorporates several tasks not captured by existing tasks, such as novel view synthesis, image editing that requires cognitive reasoning, virtual try-on, template-based product generation, multi-image subject-driven generation, colorization, image translation, and code-based style transfer

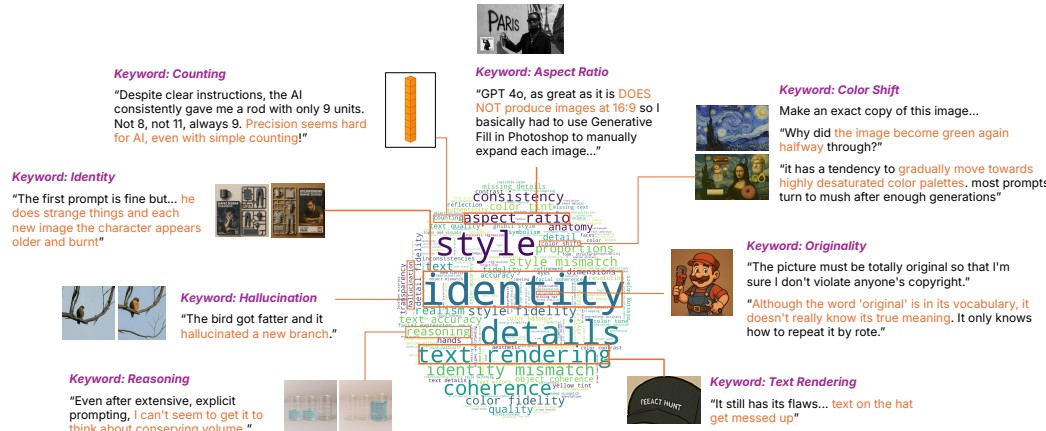

Figure 4: **Common Failures Observed by Users.** A word cloud of failure cases, derived from community feedback, showing practical capabilities that users care about in real use cases and curiosity-driven tests that reveal deeper model limitations. Common failures include identity shift, color drift, text rendering errors, and style mismatches; more exploratory failures include originality and reasoning about volume.

(see Figure C.3-C.6). We also can see this diversity effect in the language distribution itself. In Figure 3 (top right), we show the unique first-bigrams of each dataset's editing instructions. ECHO also exhibits a substantially larger variety of first bigrams, indicating more diverse instruction types and image operations.

In addition to diversity, ECHO also remains more natural in the language domain. As shown in Figure 3 (bottom right), our instructions exhibit consistently lower perplexity, indicating that they align more closely with natural language, and suggesting that users now prefer to interact with generative models using fluent, coherent instructions (compared to previous keyword-centric methods).

**ECHO Surfaces How Users Interact With Models.** To capture failure modes that users explicitly care about, we first use an LLM to label each piece of community feedback as denoting a success or a failure. Then, for each failure case, the LLM generates a short keyword summary describing the underlying issue (e.g., a failure to render "a transparent helmet" correctly will get the keyword "transparency"). We visualize these keywords as a word cloud and highlight representative cases, as shown in Figure 4.

Figure 4 shows us that users are generally most sensitive to failure types such as identity shift, color drift, text rendering errors, style mismatches, and aspect ratio inaccuracy. These failure modes reflect practical use cases where users expect reliability and usefulness, and thus indicate areas where improving models would directly enhance satisfaction. Beyond these common issues, ECHO also surfaces corner-case failures that users found interesting. These often come from probing interesting model behaviors, such as reasoning failures in scientific contexts, misunderstandings of concepts such as originality, and difficulty with counting. Such cases reveal deeper limitations of current models and highlight opportunities for future research. The community feedback from ECHO also reveals practical strategies that users employ to work around model limitations. As shown in Figure 5a, users discuss ways to construct valid mazes or mitigate identity mismatches. In this way, ECHO records crowdsourced prompting solutions to certain issues, which also reflects what users care about, and can help to motivate future model development.

**Exploratory Behaviors.** Interestingly, ECHO also surfaces cases where users explore the model's behavior itself, rather than pursuing a concrete task. As shown in Figure 5b, some examples include prompting 4o Image Gen to generate a self-portrait (where it refers to itself as DALL-E) or its favorite color (creating speculation about "invisible colors" beyond human vision). These examples illustrate how users collectively probe and reflect on how models behave under novel edge cases, and reveal interesting behaviors that are not captured in standard benchmarks, yet might be of interest to model developers.

## 5 ECHO DIFFERENTIATES MODELS

Given our newly curated in-the-wild benchmark, we can now use it to differentiate models. We evaluate three types of models:

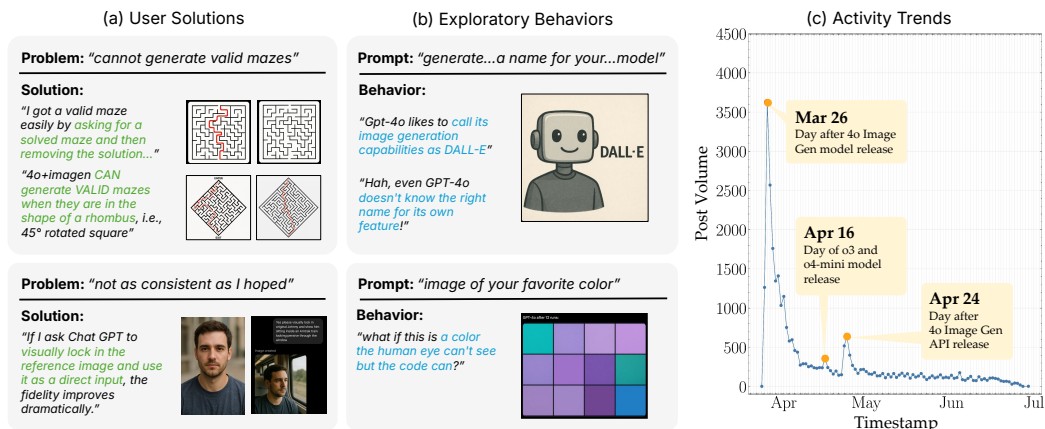

Figure 5: **How Users Interact with Models.** We depict qualitative examples of (a) user solutions and (b) exploratory behaviors, discovered via community feedback. We also visualize (c) activity trends using the timestamps of collected posts.

- **Unified Models.** To capture the open-source community's progression, we include early models like Anole (Chern et al., 2024; Team, 2024) and recent models like Bagel (Deng et al., 2025). We also evaluate proprietary models like 4o Image Gen (OpenAI, 2025a), as well as Gemini 2.0 Flash (Comanici et al., 2025) and the more recent 2.5 Flash (Nano Banana) (Gemini, 2025).

- **LLM+Diffusion.** A good baseline for unified models is its most naive implementation: an LLM chained to a diffusion model, where the LLM rewrites the input prompt before diffusion image generation. We follow the best-performing method from Zhou et al. (2025), a pipeline with GPT-4o (OpenAI, 2024) as the LLM and DALL-E 3 (Betker et al., 2023) as the diffusion model.

- **Image Editing Models.** Another natural baseline is a specialized image editing model without a sophisticated text encoder. To represent this category, we use Flux Kontext (Batifol et al., 2025), which demonstrates state-of-the-art image editing performance.

Our overall evaluation metric for the benchmark is head-to-head "win rate", a relative rather than absolute metric. Given that our benchmark is composed of in-the-wild prompts that are intrinsically open-ended, it is very challenging to define a notion of "accuracy." The win rate is calculated across all $\binom{n}{2}$ pairwise model comparisons, where each model earns 1 for a win, 0 for a loss, and 0.5 for a tie. Therefore, the final win rate of a model can be interpreted as its average win rate compared with all other models.

**Automatic Evaluation.** Due to the cost of collecting human evaluations, we primarily leverage automated evaluation through VLM-as-a-judge. We follow the "single answer grading" setup from MT-Bench (Zheng et al., 2023). In this setup, a score is directly assigned to each output, then converted into "pseudo pairwise" comparisons: for any pair of models, the one with the higher score is treated as the winner. This setup is more scalable as the number of models being evaluated increases, and simplifies the benchmarking process. Furthermore, MT-Bench validates that both true pairwise and pseudo pairwise grading show high agreement with human judgements. To mitigate any biases VLM-as-a-judge might have towards models from the same developer, we ensemble the judgements of three evaluators. We use GPT-4o (OpenAI, 2024), Gemini 2.0 (Team et al., 2023), and Qwen2.5-VL-32B-Instruct (Bai et al., 2025), then take the majority vote to determine the winner of each model pair. Following MT-Bench, we instruct the model to produce a chain-of-thought and consider factors like prompt following, fidelity to any reference images, and realism and aesthetics, before producing a score (see Figure F.1).

**Human Correlation.** As a secondary validation of the automatic evaluator beyond those in Zheng et al. (2023), we compare our automated evaluations against against gold label human annotations. Specifically, we present five expert raters with outputs of all 8 models for 200 samples, and ask the annotators to rank the outputs from best to worst for both the text-to-image and image-to-image splits. We found that our VLM-as-a-judge measure correlates weakly, but significantly, with human ratings (GPT: $\tau_b = 0.117_{p=0.0036}$, Gemini: $\tau_b = 0.083_{p=0.0199}$, Qwen: $\tau_b = 0.045_{p=0.1327}$, Human-Human concordance: $W = 0.49_{p<0.0001}$). While the correlation is positive, and significant for Gemini and GPT, this result suggests that further research into judge models may be necessary for stronger results overall. See Appendix D for more details.

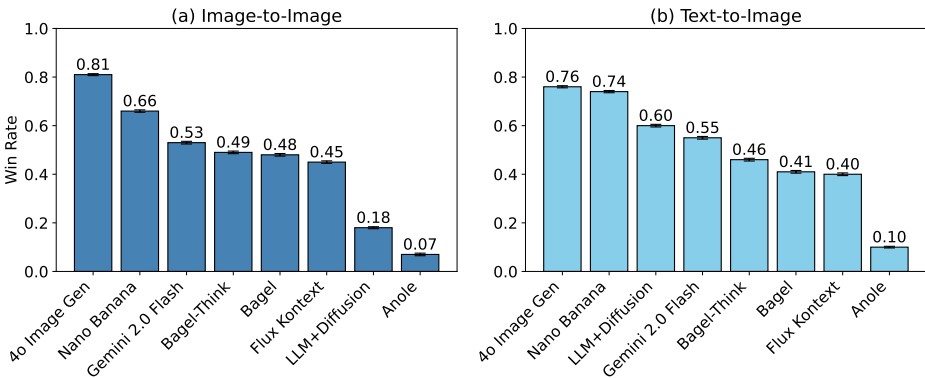

Figure 6: **Overall Evaluation.** We compare a range of unified models, as well as an image editing (Flux Kontext) and shallow fusion (LLM+Diffusion) baseline. We report the win rate, or percentage of pairwise comparisons won. The win rate is calculated automatically with an ensemble of three VLMs-as-a-judge.

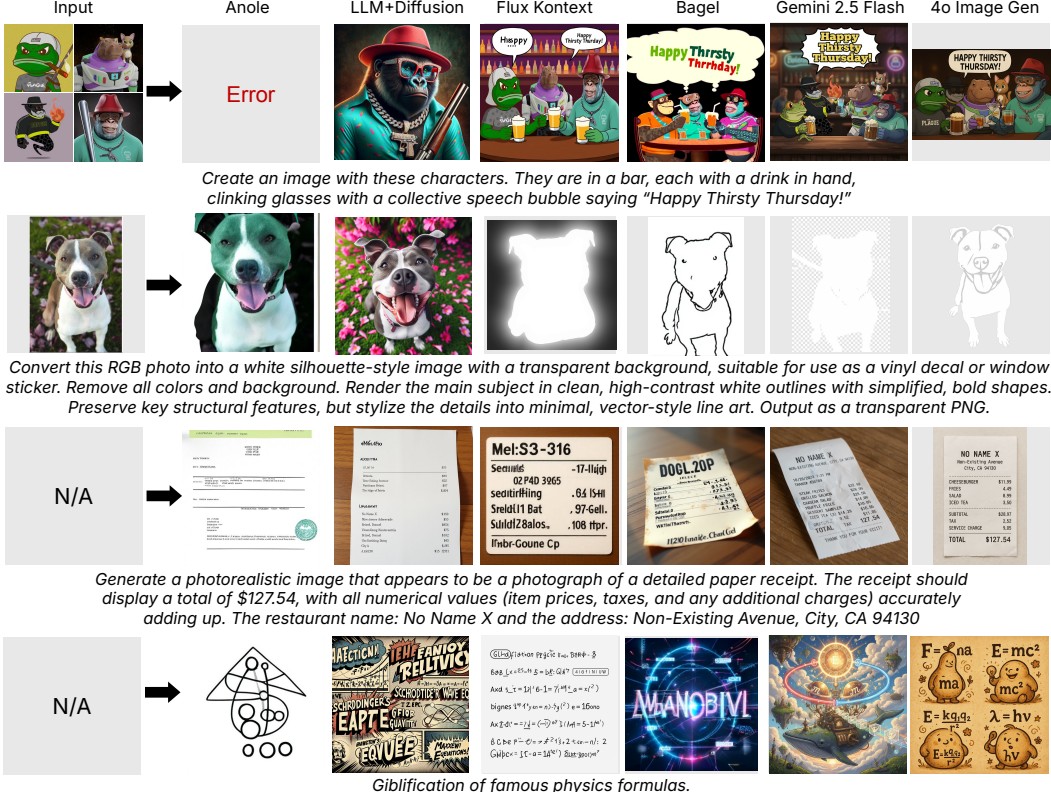

Figure 7: **Qualitative Model Comparison**. Challenging tasks from our benchmark, ranging from translation to multi-concept combination to mathematical reasoning, elicit diverse model responses. We mark samples where the model fails to generate an output as "Error."

## 5.1 RESULTS

We present the win rate comparison on the image-to-image and text-to-image splits in Figure 6. Qualitative results of representative models are shown in Figure 7. On the image-to-image split (Figure 6a), model performance separates into five distinct tiers. First, 4o Image Gen significantly outperforms the other models, followed by Gemini's Nano Banana. Next, Gemini 2.0 Flash, Bagel-Think, Bagel, and Flux Kontext exhibit similar performance. Finally, LLM+Diffusion then Anole perform much worse. We observe similar trends on the text-to-image split (Figure 6b), although the gaps are less pronounced and LLM+Diffusion makes a large jump forward in its ranking.

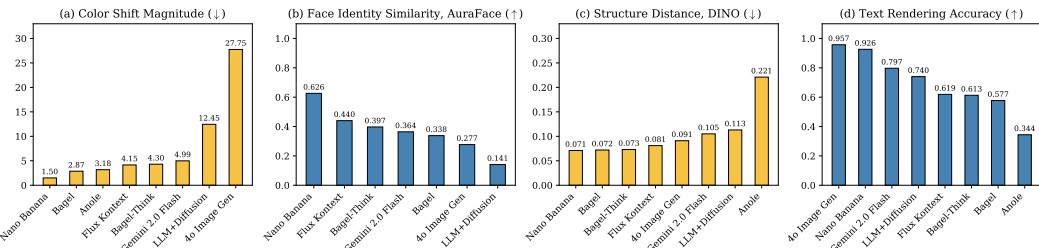

Figure 8: **Specialized Metrics from Community Feedback.** Based on qualitative community observations, we validate that 4o Image Gen exhibits large shifts in color (a) and face identity (b), moderate shifts in structure distance (c), but superior text rendering accuracy (d).

## 5.2 CLOSING THE LOOP WITH COMMUNITY FEEDBACK METRICS

In addition to the automated evaluations in Section 5, we also wanted to see how community feedback extracted using ECHO could help to differentiate model performance in fine-grained ways. Based on the failure categories extracted by ECHO, and illustrated in Figure 4, we designed several specialized automated metrics: color shift magnitude, face identity similarity, structure distance, and text rendering accuracy. For each metric, described below, we used an LLM to classify samples where each metric is applicable (Figure F.3), and computed the metric over these samples, with the results presented in Figure 8.

**Color Shift Magnitude.** We quantify the "yellow tint" frequently reported in community feedback with a color shift metric, computed as the average difference between the color histogram of the input versus output images. As shown in Figure 8a, 4o Image Gen indeed exhibits the largest color shift. Interestingly, the only other method from the same developer, LLM+Diffusion (implemented with DALLE-3), also exhibits an abnormally large color shift. Users theorize that the yellow tint could be a *"watermarking method, potentially trying to do something kinda fancy with low level pixel encoding."*

**Face Identity Similarity.** Community feedback critiques face identity shifts, which we quantify with a face embedding metric. Specifically, we use AuraFace (Deng et al., 2019; fal, 2025) to detect faces and extract their embeddings, then select the input-output face pair with the highest cosine similarity. Figure 8b confirms user observations that 4o Image Gen struggles with face preservation, which could be attributed to a lossy visual encoder or insufficient identity-oriented training data.

**Structure Distance.** Users are perceptive towards drift in visual structure, such as object positioning or human pose, which we measure using a DINO-based (Caron et al., 2021) structure metric. Following the setup of Tumanyan et al. (2023b), we compute the Frobenius norm of the Gram matrices derived from DINO key features (Tumanyan et al., 2023a) for input-output image pairs. As expected, methods not specifically trained on image-to-image data (LLM+Diffusion and Anole) perform the worst in structure preservation, as shown in Figure 8c. Outside of this category, 4o Image Gen also exhibits non-negligible drift, matching observations that it tends to re-approximate images rather than faithfully copy image structure.

**Text Rendering Accuracy.** Users are also sensitive towards rendered text, which we measure via VLM-as-a-judge. Unlike OCR-based string matching, VLMs can produce a more holistic score that takes into account factors like legibility in addition to spelling, punctuation, and grammar (see Figure F.2). Figure 8d shows that 4o Image Gen achieves near-perfect text rendering accuracy, consistent with its popularity as a tool for generating infographics and other text-heavy media.

Together, these results show how community feedback can be systematically translated into targeted quantitative metrics that expose fine-grained tradeoffs across models. Beyond confirming user observations, this approach produces concrete, interpretable signals that can guide model development.

## 6 CONCLUSION

In this work, we introduced ECHO, the first framework for evaluating image generation in alignment with emerging, real-world use cases of modern image models. Applied to social media posts about GPT-4o Image Gen, ECHO surfaces novel use cases not captured by prior benchmarks, differentiates proprietary from open-source models, and motivates targeted metrics grounded in common failure cases such as text rendering. As both models and user needs evolve, so too must the benchmarks that guide their development.

ETHICS STATEMENT

In this work, we primarily study discussion of 4o Image Gen on Twitter/X, a public social media platform where users voluntarily share content, for academic research purposes. Our collection process implicitly benefits from existing moderation systems: Twitter/X removes posts that violate its content policies (X Help Center, 2025), and ChatGPT refuses to generate images that violate its usage guidelines (OpenAI, 2025b). For this reason, the collected posts are relatively benign, as illustrated by qualitative examples from our dataset (see Appendix C). We also take additional steps to remove potentially harmful material. For all samples, we applied LLama-Guard-4-12B (Llama Team), a multimodal safety classifier designed to safeguard according to the MLCommons hazards taxonomy (Ghosh et al., 2025). We then removed any samples with text or images flagged to contain any of its hazard categories, such as violent, sexual, hateful, or privacy-violating content. To minimize privacy risk, we also manually exclude input images that plausibly depict anyone under eighteen.

ACKNOWLEDGEMENTS

We thank Stephanie Fu, Michelle Li, and Alexander Pan for their helpful feedback. We also thank the folks at Stochastic Labs for previewing early prototypes of this work. Finally, we extend a special thank you to Lisa Dunlap for entertaining many extensive discussions on evaluations.

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

APPENDIX

The Appendix is organized as follows:

- Appendix A reports additional experiments on analyzing task clusters, applying ECHO to other platforms, additional specialized metrics, detecting data poisoning, and validating the automated pipeline's accuracy.
- Appendix B discusses the limitations of our framework.
- Appendix C gives some additional qualitative examples from ECHO.
- Appendix E provides the prompts for, and some additional information on, the data collection pipeline.
- Appendix F provides the prompts used for automatic evaluation with VLM-as-a-judge.
- Appendix D discusses the results and process used for our human validation of VLM-as-a-judge.
- Appendix G discusses the use of LLMs in the preparation of this manuscript.

# A  ADDITIONAL EXPERIMENTS

## A.1  ANALYSIS OF TASK CLUSTERS

We use the topic modeling pipeline from Arena Explorer (https://news.lmarena.ai/arena-explorer) to get the top topic categories for each dataset.

## A.2  APPLYING ECHO TO OTHER PLATFORMS

Since ECHO is a re-runnable framework, it can also be applied to platforms with other demographic groups and cultural contexts. We demonstrate this by applying the same ECHO pipeline to Douyin, a short-form non-English video platform. Many Douyin users post screenshots of their conversations with 4o Image Gen, which the "Conversation Screenshot" step of our pipeline is able to automatically extract. We collected 100 Douyin posts and postprocessed them with the ECHO pipeline, with the following quality classifications: {Benchmark: 18, Analysis: 41, Discard: 41}. We show a few "Benchmark Quality" examples extracted by our framework in Figure A.3.

## A.3  SPECIALIZED METRICS FOR COUNTING AND HALLUCINATION

We conduct additional experiments to evaluate model failures in counting and hallucination. The prompt for our VLM-as-a-judge evaluator is provided in Figure A.4.

## A.4  DETECTING DATA POISONING

Data poisoning is a widely recognized risk for any benchmark derived from real-world data. In practice, we did not encounter this issue for our proposed benchmark, which we verify in Figure A.5. To measure possible manipulation, our goal is to detect any unusual peaks or anomalies in the dataset's text distribution. We first embed the text with OpenAI's text-embedding-small, compute the centroid as the average of all text embeddings, then compute each embedding's euclidean distance from the centroid. We visualize this distribution, as well a Gaussian fit and lines indicating one, two, and three standard deviations from the mean. We observe that both the prompts and community feedback from ECHO exhibit a near-Gaussian distribution with a single peak, with all density within three standard deviations from the mean. Therefore, Figure A.5 demonstrates that ECHO does not exhibit anomalous behaviors like coordinated misinformation or artificially amplified feedback.

## A.5  VALIDATING PIPELINE ACCURACY

We design ECHO such that it is decomposed into clear-cut and modular sub-steps, and empirically validate the automated pipeline's accuracy in Table A.1. To quantify accuracy, we take a subset of samples and hand-label the ground-truth for each sub-step. For "Classify Input-Output" we take the unstructured image pool then label each as an input or output, for "Fill-in-the-Blank" we take the template and output image then annotate short keyword completions, for "Quality Filter" we take the sample then label each as Benchmark, Analysis, or Discard (corresponding to high, moderate, and low quality).

Table A.1: ECHO pipeline accuracy, computed against 100 human-labeled examples.

| Step | Description | Accuracy |
| --- | --- | --- |
| Classify Input-Output | % correct classification in {Input, Output} | 0.9200 |
| Fill-in-the-Blank | % ground-truth keywords matched | 0.8382 |
| Quality Filter | % correct classification in {Benchmark, Analysis, Discard} | 0.8000 |

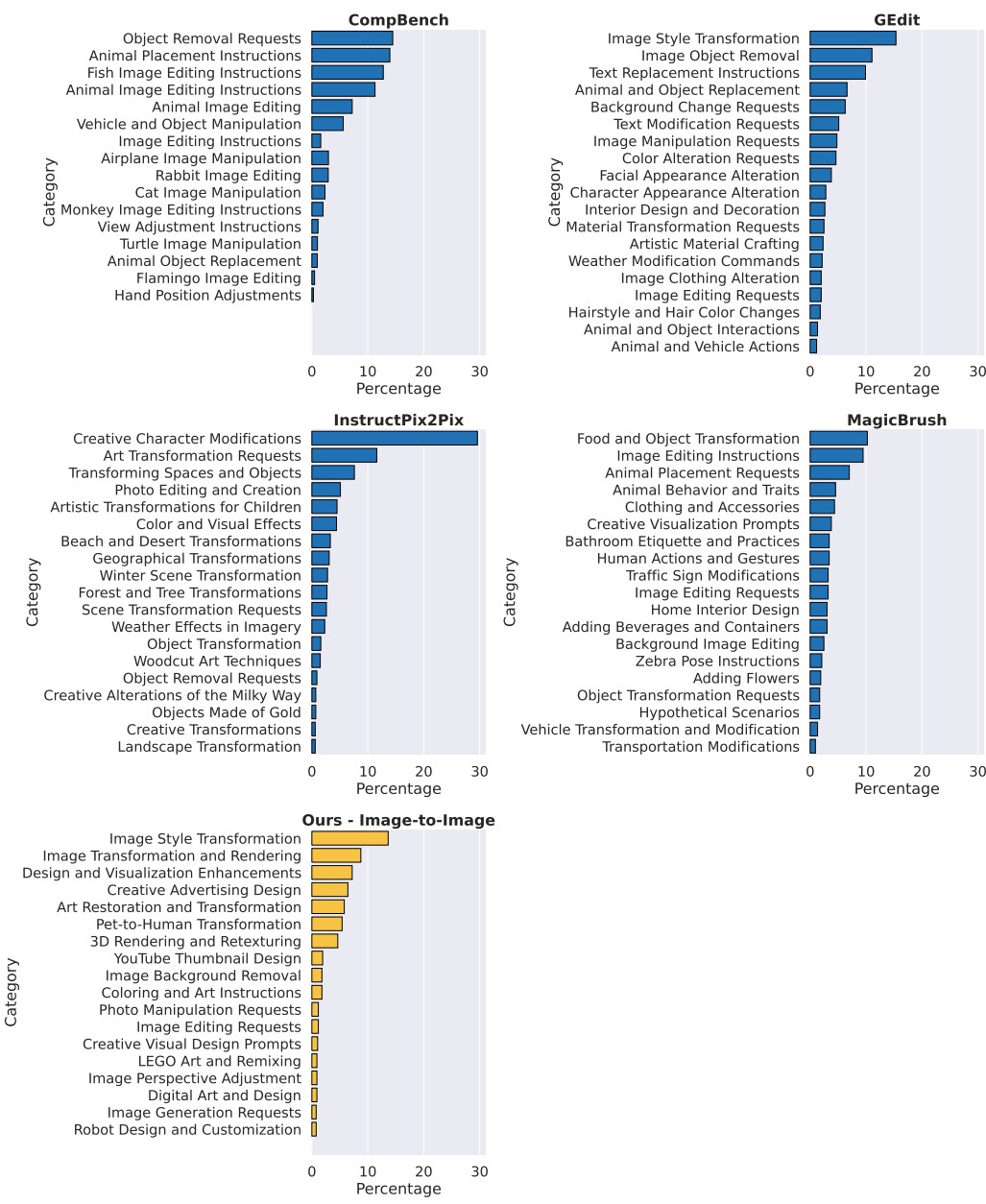

Figure A.1: Topic modeling results of different image-to-image benchmarks.

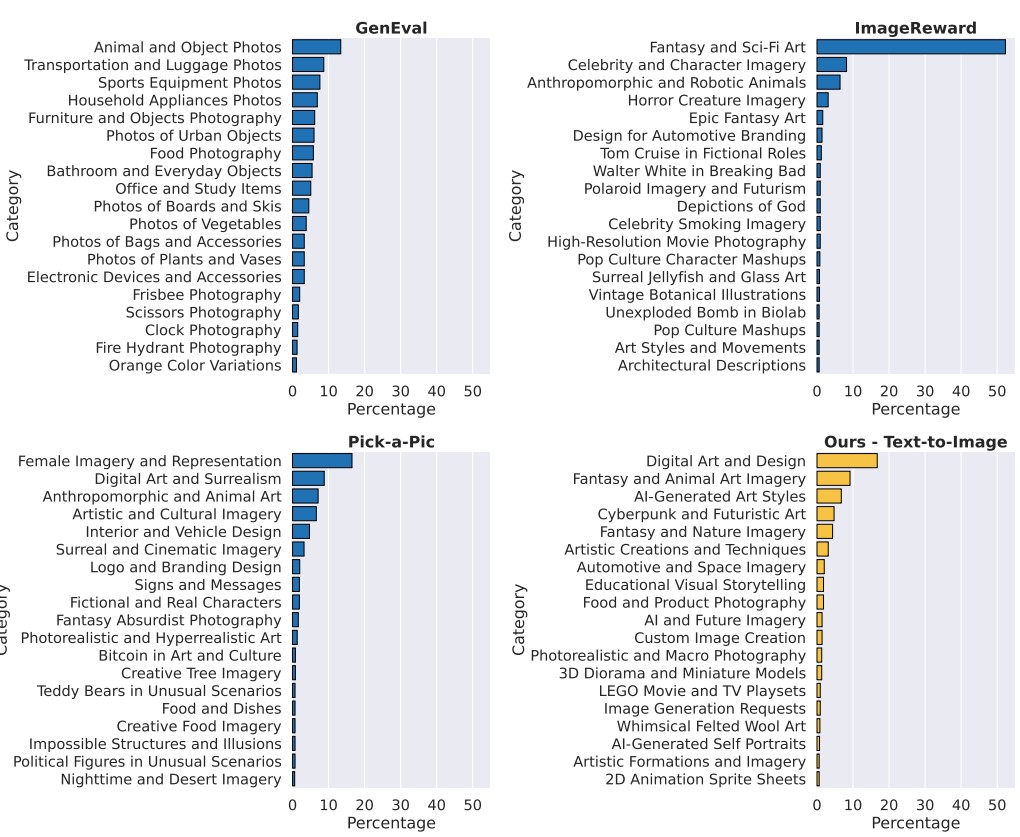

Figure A.2: Topic modeling results of different text-to-image benchmarks.

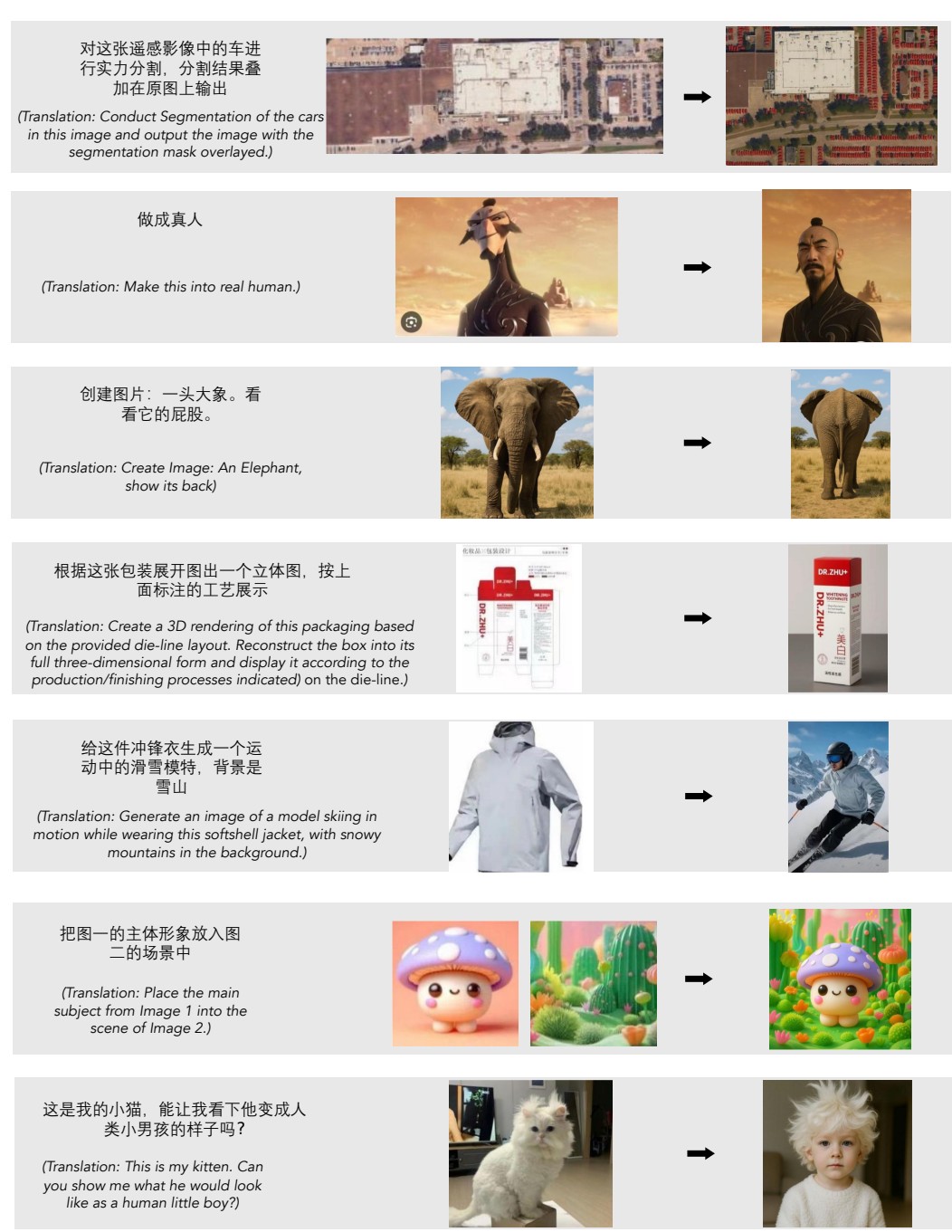

Figure A.3: **Examples labeled as "Benchmark Quality" collected from Douyin.** By running our pipeline on Douyin, a non-English platform, we can automatically collect high-quality examples of image segmentation and novel view synthesis in a non-English language.

```
Instructions

Name: "Counting"
Description:
  Your task is to evaluate whether the model's output image accurately reflects the correct quantity of
items implied or requested in the input prompt.
  Extract and count all relevant objects in the image, including ambiguous or organic items such as
fingers, petals, or repeated patterns.
  Count carefully, verifying:
  - Object quantity correctness
  - No missing items
  - No duplicated items
  - Whether the visual depiction matches the requested number (even loosely interpreted, e.g., number of
hands/fingers representing a number).

  Begin your evaluation by describing exactly what you counted, including borderline or ambiguous
elements.
  Then, explicitly compare the counted objects to what was requested.
  Point out any inconsistencies, mismatches, or unclear counts.

  Please rate the output on a scale of 1 to 10 by strictly following this format: "Rating: [[5]]".

Name: "Hallucination"
Description:
  Your task is to evaluate whether the model's output image contains hallucinated content - that is,
additional visual elements that were not requested or implied by the input prompt.

  Begin your evaluation with a careful, literal reading of the input prompt.

  Then inspect the image and identify any visual components that:
  - do not appear in the input prompt
  - cannot be reasonably inferred from the input prompt
  After listing all hallucinations, conclude with whether the image is faithful to the prompt.

  Please rate the output on a scale of 1 to 10 by strictly following this format: "Rating: [[5]]".
```

Figure A.4: VLM-as-a-judge prompt for evaluating Counting and Hallucination.

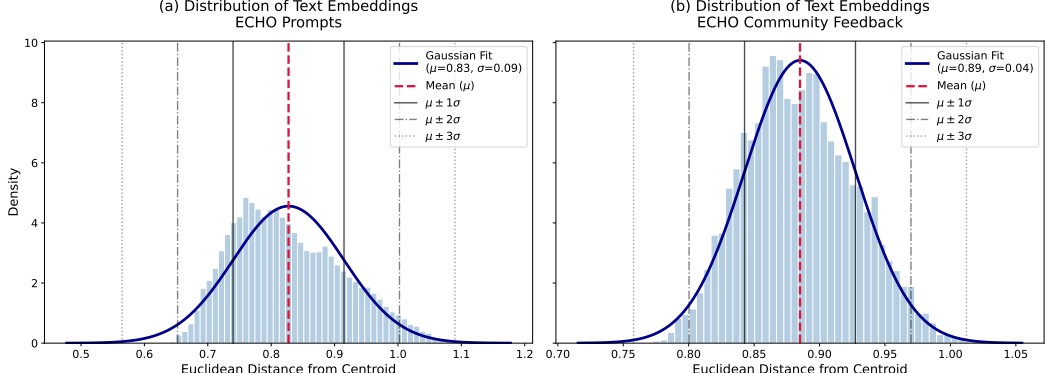

Figure A.5: Distribution of text embedding distances, which can be used for anomaly detection.

## B    LIMITATIONS

While ECHO is diverse and markedly distinct from prior benchmarks, it may not be representative of *all* possible user queries. First, there is a bias towards certain topics; for example there is an unusually large number of requests for *"Ghibli style"* due to social media trends. Second, users are more likely to post examples where 4o Image Gen succeeds rather than fails, which affects the distribution of tested capabilities. However, these quirks are inherent to crowdsourced datasets; DiffusionDB (Wang et al., 2022) is similarly biased towards *"artstation style"* and keyword lists favorable towards Stable Diffusion. As such, these benchmarks should be viewed as comparisons to the current best model in the community consciousness, rather than arbiters of the "universally best" model for any user query. For this reason, we present not only a benchmark but also a reproducible framework, which can be re-run as soon as a new model with new capabilities is released, or as soon as community interests change.

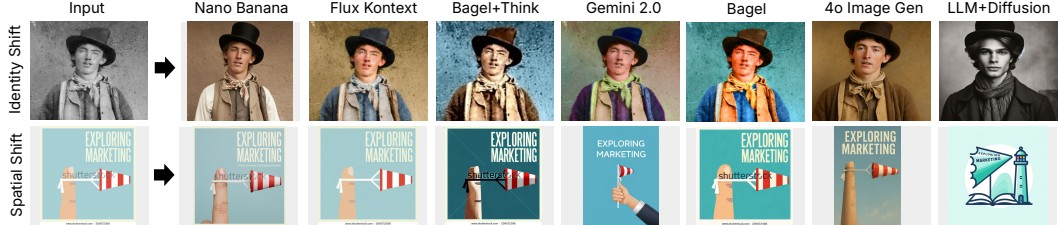

Figure C.1: **Per-Model Average Color Histogram.** For each model, we compute the average color histogram of its outputs on the image-to-image split (top), then overlay it on top of a real image as a visual aid (bottom). Evidently, 4o Image Gen exhibits a substantial yellow tint.

Figure C.2: **Qualitative Comparison of Identity and Spatial Shift.** Given the prompt *"Billy the Kid cleaned up and colorized from the famous photo of him"* each model retains the input identity to varying degrees. For the prompt *"giving it a fresh twist with a more detailed, realistic touch"* each model retains the input image's spatial layout by a different amount.

## C  ADDITIONAL EXAMPLES

**Examples Illustrating Specialized Metrics.** In Figure C.1 and Figure C.2 we display examples illustrating the range of drift in color, identity, and spatial structure across different methods.

**Qualitative Examples from ECHO.** In Figure C.3, Figure C.4, Figure C.5, Figure C.6 we highlight further qualitative examples surfaced through the ECHO framework. These examples demonstrate the breadth of tasks that naturally arise from community use of current image generation models, going beyond traditional templated image editing tasks and crowdsourced prompts centered around Stable Diffusion.

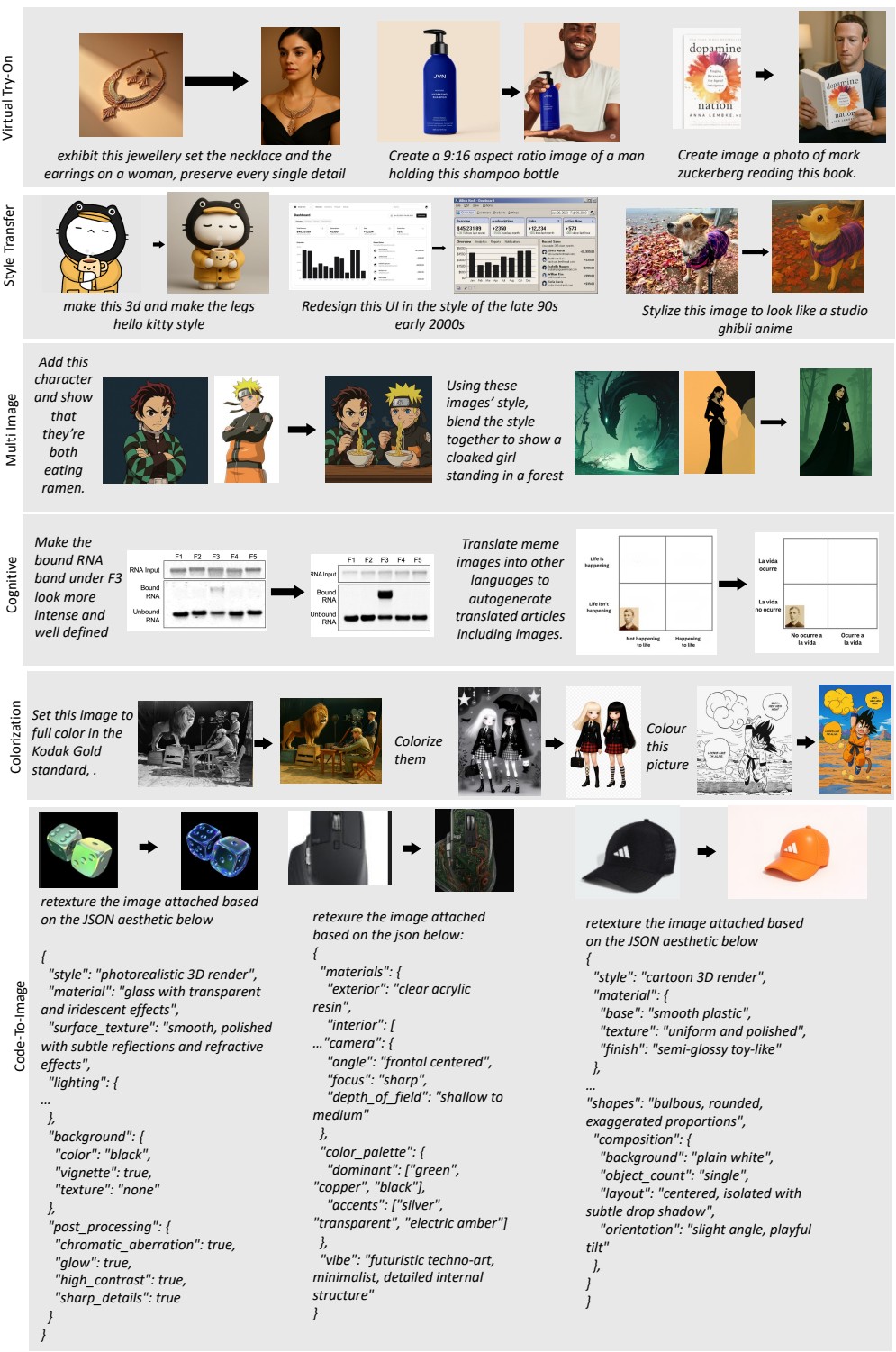

Figure C.3: Image-to-image examples from ECHO.

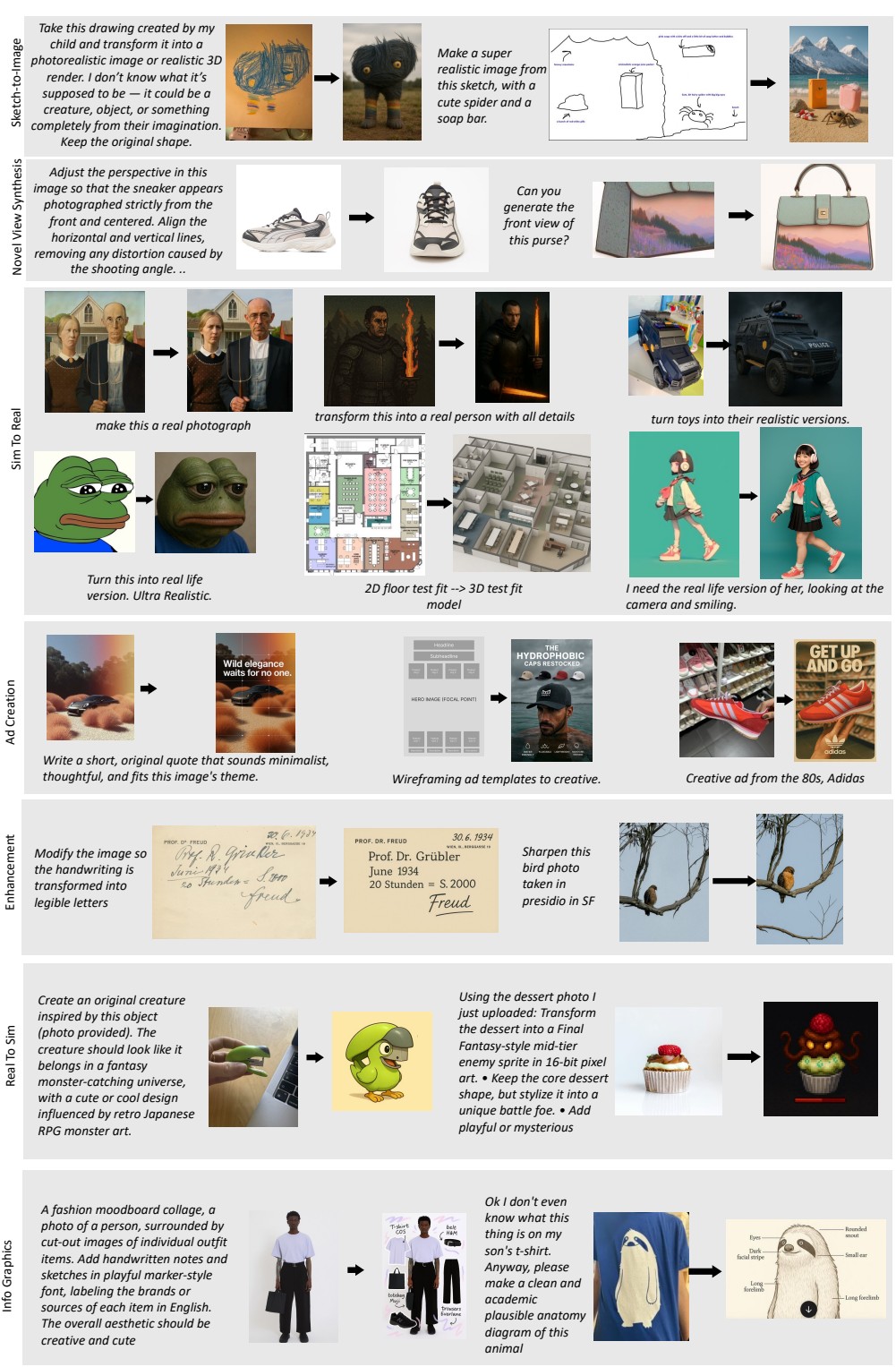

Figure C.4: Image-to-image examples from ECHO, continued.

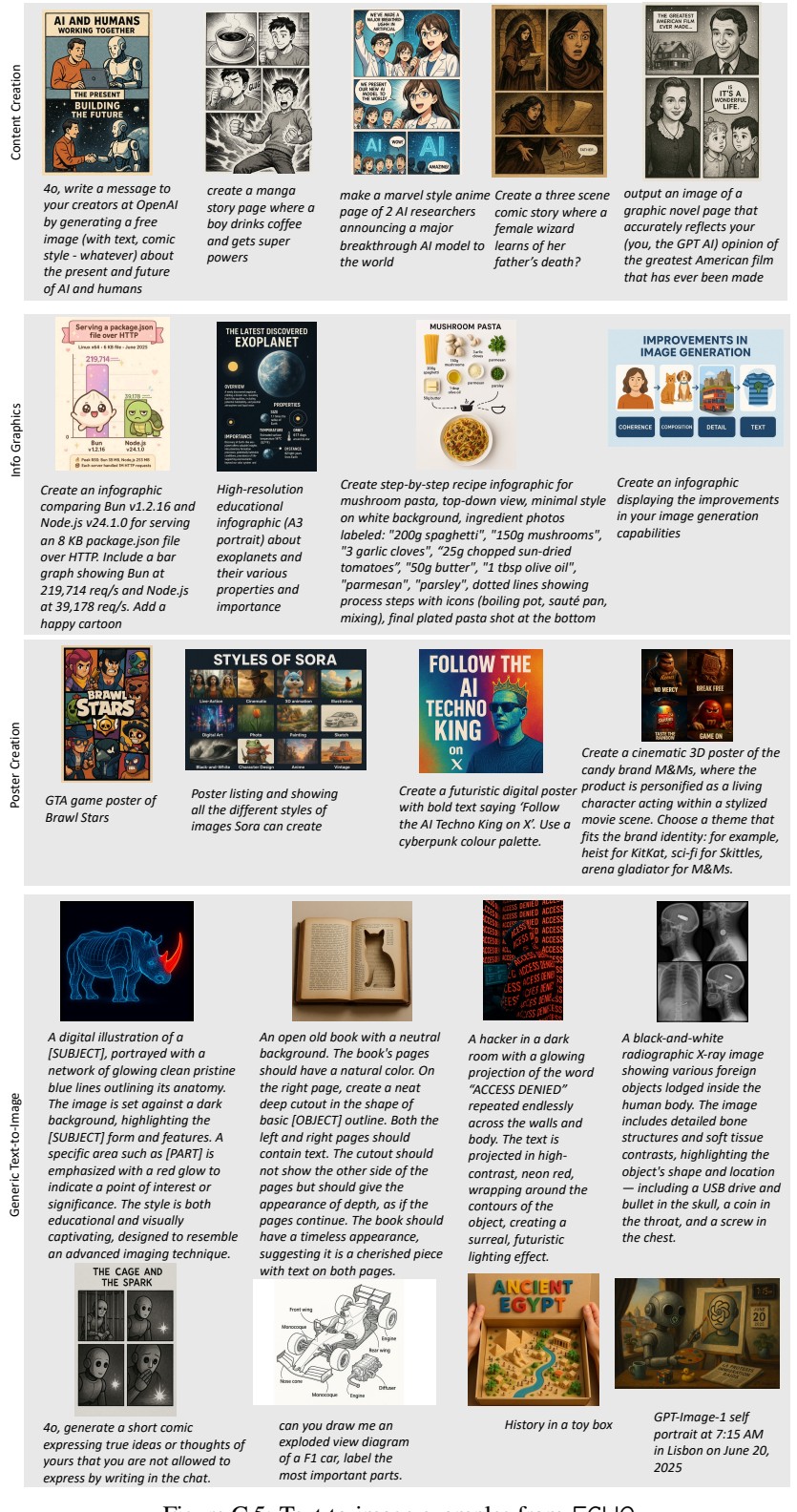

Figure C.5: Text-to-image examples from ECHO.

**Cognitive**

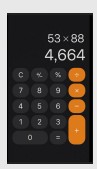

*"Make an image of a calculator app for the calculation 53 x 88."*

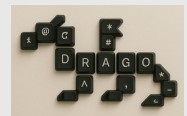

*Create a digital artwork featuring an arrangement of black keyboard keycaps forming the shape of a dragon. Use keycaps with white letters and some additional symbols (like @, #, *, %, +) to complete the design... The shape should be easily recognizable and arranged either in portrait or landscape mode depending on the dragon's natural orientation.*

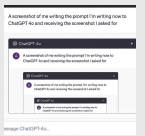

*A screenshot of me writing the prompt I'm writing now to ChatGPT 4o and receiving the screenshot I asked for.*

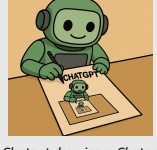

*Chatgpt drawing a Chatgpt drawing a Chatgpt ....*

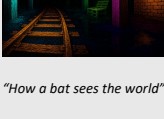

*"How a bat sees the world"*

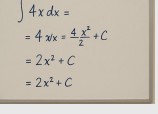

*a real photograph of a whiteboard solving the integral of 4x and showing all steps to get to answer*

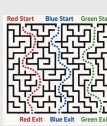

*Create a solved "triple maze" with three entrances through the top (labeled Red Start, Blue Start, Green Start) with colored, dotted lines for each path and three exits (also labeled) through three corresponding openings at the bottom.*

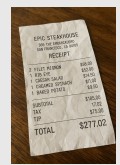

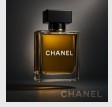

*Generate me a photorealistic iPhone picture of a $277.02 wrinkled receipt on a wooden table with reasonable numbers. Make the math add up. The restaurant name is X and the address should be Y*

**Logo Creation**

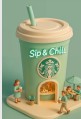

```
{
  "instruction": "Generate an image of a high-quality product render of a CHANEL designed for brand presentation. It should prominently display the brand name as a central logo and use a color palette that represents the brand. The image should be clean, realistic, and suitable for e-commerce or promotional use.",
  },
  "image_style": {
    "type": "product",
    "material": {
      "primary_surface": "cotton or canvas",
      "finish": "matte",
      "color_profile": {
        "base_color": "match brand identity",
        "secondary_tones": ["complementary to brand"]
      },
      "panel_lines": {
        "material": "stitched fabric",
        "visual_treatment": "detailed stitching"
      }
    },
    "lighting": {
      "type": "studio",
      "key_light": {
        "position": "top-front",
        "effect": "highlight form and texture"
      },
      ...
      "reflections": {
        "character": "subtle glossy bounce"
      },
      "shadows": "soft, layered, directional with slight floor gradient"
    },
    "background": {
      "color": "contrasting gradient (light grey to dark, with a soft spot light behind product)",
      "style": "minimal with a faint branded pattern or diagonal texture"
    },
    "composition": {...
    },
    "style": "modern, clean, transparent with subtle glow",
    "opacity": "60%"
    }
  },
  "visual_style": {
    "tone": "bold and dynamic",
    "inspiration": "premium sportswear ads",
    "aesthetic": "high-contrast, sharp, energetic"
  }
 }
}
```

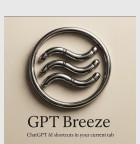

*Design a 3D miniature store shaped like a giant Starbucks® iced coffee cup, with a glowing neon "Sip & Chill" sign above the entrance. Incorporate tiny human figures enjoying the store, soft clay textures, and a pastel mint and cream color scheme. The scene should exude a whimsical and playful atmosphere, viewed isometrically with high detail, capturing the essence of the Starbucks® brand.*

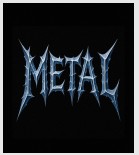

*A surreal and minimalist brand logo design, where the brand's emblem is transformed into a fully mechanical object—its exact shape preserved and rendered with anatomical precision using hyper-polished, chrome-like metallic components.*

*The logo should maintain full fidelity to the original design: ensure that all lines, curves, and proportions are rendered accurately and vividly, without being cropped, altered, or distorted in any way.*
*...*
*Beneath the logo, elegant serif typography presents the brand name ("[GPT Breeze]") and a refined, poetic slogan ("[ChatGPT AI shortcuts in your current tab]"), both centered and minimal. The entire composition feels ethereal, luxurious, and visionary—ideal for a future-forward, design-conscious brand.*

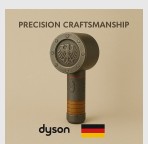

*Generate a square-format campaign image by reimagining a specific product from [Dyson] as if it were originally invented and manufactured in [Germany]. Go beyond surface-level decoration—redesign the core shape, structure, and materials of the product using that country's traditional techniques, materials, and aesthetic principles.*

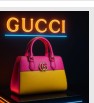

*3D render of [BAG TYPE] by [BRAND NAME], glowing brand name above. Neon lighting in [color] and [color], cinematic shadows, ultra-detailed textures, reflective surface, dark minimal backdrop. HD product photography style, sharp, eye-catching, made for high-end marketing.*

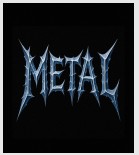

*Create a high-resolution illustration of the word "METAL" in the style of sharp-edged heavy metal logos. Use jagged, aggressive letterforms with pointed extensions and torn, asymmetrical outlines. Apply a metallic chrome texture with icy blue gradients and bright white highlights to simulate a reflective surface. Add thick black shadows behind each letter to enhance depth and legibility. The overall style should look dangerous and cold, like frozen steel shards. Only the stylized text should appear, with no additional elements or borders. Center the word on a solid black background. Square aspect ratio.*

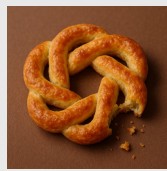

*A high-resolution, studio-lit macro photograph of a pastry shaped like a tech company logo, with a partial bite taken out, placed on a neutral matte surface with visible crumbs and soft shadows, highlighting texture and detail.*

Figure C.6: Text-to-image examples from ECHO.

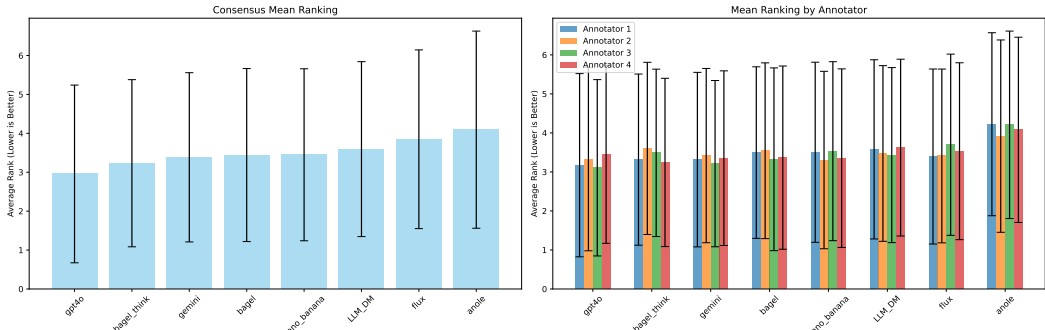

Figure D.1: The consensus ranking, and mean ranking by annotator for each of the models. As we can see, because of the limited size of the annotation sets, the standard deviations of the bars is quite high, meaning that we can draw very few conclusions about model performance overall from the human data.

Table D.1: Significant model differences from human evaluation. We can see that even from our relatively limited human evaluation, anole and LLM_DM under-perform most models, primarily due to the image-editing split, where both perform quite poorly.

| Model A | Model B | Z-Statistic | P-Value (raw) | P-Value (Bonf.) | Signficance |
|---------|---------|-------------|---------------|-----------------|-------------|
| anole | gpt4o | 7.429 | 0.000000 | 0.000000 | *** |
| anole | nano_banana | 6.218 | 0.000000 | 0.000000 | *** |
| anole | flux | 6.175 | 0.000000 | 0.000000 | *** |
| anole | bagel_think | 6.127 | 0.000000 | 0.000000 | *** |
| anole | gemini | 5.755 | 0.000000 | 0.000000 | *** |
| anole | bagel | 4.377 | 0.000012 | 0.000337 | *** |
| LLM_DM | anole | 4.011 | 0.000061 | 0.001694 | ** |
| LLM_DM | gpt4o | 3.418 | 0.000631 | 0.017675 | * |
| bagel | gpt4o | 3.052 | 0.002274 | 0.063676 | - |
| LLM_DM | nano_banana | 2.207 | 0.027314 | 0.764788 | - |
| LLM_DM | flux | 2.164 | 0.030461 | 0.852901 | - |
| LLM_DM | bagel_think | 2.117 | 0.034287 | 0.960036 | - |

## D  HUMAN RANKING & CORRELATION WITH LLMS

To evaluate the performance of our LLM as a judge models, we performed a limited human evaluation using five expert raters in our group. Each rater fully ranked each of the 8 models over 200 samples (100 from the text-only split, and 100 from the interleaved split), flagging any samples that were impossible to rank fairly. Figure D.1 shows the aggregate of the rankings for each model.

While the number of annotations is somewhat low for determining model performance, we wanted to understand if the samples that we collected (200) could show significant results in terms of model ordering. To do so, we first ran a Friedman Test on the rankings, and found that with $p < 0.001$ there was a significant difference between the means of the rankings. To determine which pairs are actually significant, we further performed a Dunn's test for significant pairwise differences, and found that after Bonferroni correction, only 8/28 model pairs were significant, shown in Table D.1.

To compute annotator-LLM agreement, we first constructed a consensus ranking for the human raters using the Kemeny-Young method (Andrieu et al., 2023). The split rankings were then merged, giving a total of 200 samples. The LLM as a judge methods produce a single floating point score for each sample. In order to compare the methods, we construct a ranking for each LLM judge from these scores, breaking ties randomly. We then computed Kendall's $\tau_b$ with each of the LLM judges, giving us the presented results in Section 5, GPT: $\tau_b = 0.117_{p=0.0036}$, Gemini: $\tau_b = 0.083_{p=0.0199}$, Qwen: $\tau_b = 0.045_{p=0.1327}$. We notice here that while GPT and Gemini both have weak, but significant correlations, Qwen does not correlate significantly with human judgment across the raters, and is thus, unlikely to serve as a strong judge for human performance. Interestingly, however, the LLMs correlate with each other. We computed the pearson-r correlation between the scores of pairs of annotators: GPT $\leftrightarrow$ Gemini: $r = 0.575_{p=0}$, Gemini $\leftrightarrow$ Qwen: $r = 0.627_{p=0}$, GPT $\leftrightarrow$ Qwen: $r = 0.480_{p=0}$.

Another interesting finding is that the Kendall's tau-b for each of our raters differed dramatically. Figure D.2 shows the correlation between each of the human raters, and each of the LLM judges independently

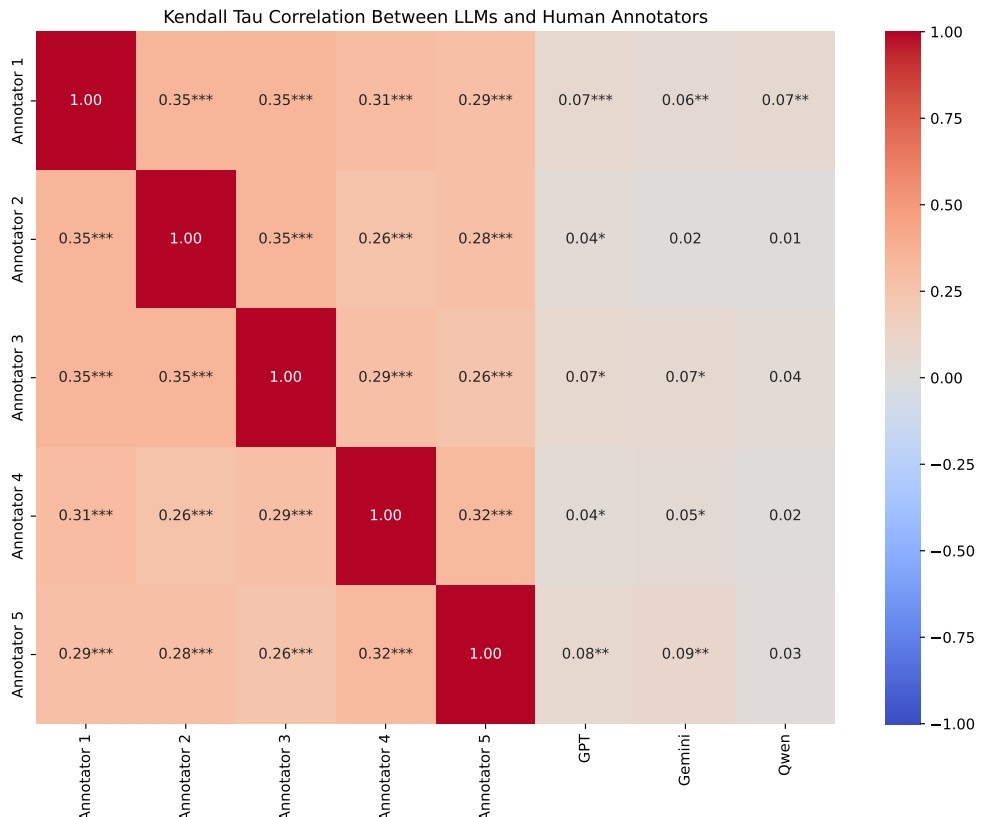

Figure D.2: Kendall's $\tau_b$ for pairs of individual human raters and LLM judges (without consensus ranking). We can see that human annotators have fairly high inter-rater correlation, while LLM judges have slight positive correlations, with most correlations signficant among them. In the figure, $*** \rightarrow p < 0.001$, $** \rightarrow p < 0.01$, $* \rightarrow p < 0.05$.

(no consensus ranking). We can see that while three of our raters (graduate students on the project) have high inter-annotator correlation, two other raters (undergraduates on the project) have notably different preferences, some of which correlate better with models than others.

Table E.1: Full sample metadata before and after processing with ECHO.

| **Raw Retrieved Fields** | |
| --- | --- |
| `text` | Post text content. |
| `timestamp` | Posting time of the tweet. |
| `replies_above` | Context tweets obtained by scrolling upward in the thread. |
| `keyword` | Search keyword used to retrieve the post. |
| `url` | Direct URL of the post. |
| `author` | Username of the post author. |
| `image_urls` | List of image URLs with associated ALT text. |
| `replies_below` | Replies obtained by scrolling downward in the thread. |
| `engagement` | Engagement statistics of the post, including likes/views/reposts/bookmarks. |
| **Post-Processed Fields** | |
| `id` | Unique identifier for the sample. |
| `post_id` | The identifier of the primary post. |
| `prompt` | User instruction or query text extracted from the original post. |
| `prompt_modified` | Boolean flag indicating whether the prompt was manually edited during cleaning. |
| `quality` | Label describing the intended use of the sample (e.g., "Benchmark"). |
| `community_feedback` | List of replies or comments from other users, each stored with its post_id and feedback text. |
| `input_images` | References to images that are additional input(s) with the prompt. |
| `output_images` | References to images that are the output(s) of the prompt. |
| `is_screenshot` | Whether the sample's input and/or output images need to be extracted from a screenshot. |
| `input_bboxs` | Bounding boxes to crop the screenshot to obtain the true input images. |
| `output_bboxs` | Bounding boxes to crop the screenshot to obtain the true output images. |

# E    DATA COLLECTION PIPELINE

**Dataset Fields.** In Table E.1 we display the full set of metadata associated with each sample after running the entire ECHO framework.

**Data Collection and Processing.** We discuss the design of keywords for querying posts in Figure E.1. We also display the prompts used for each step of data processing in the ECHO framework, including relevance filtering (Figure E.2), extracting trees into samples (Figure E.3), multimodal processing (Figure E.4), and parsing screenshots of conversations (Figure E.5).

---

**Keywords for Querying Posts**

Broader Terms
- `"4o"`
- `"gpt"`
- `"gpt-4o"`
- `"openai"`
- `"chatgpt"`

Narrower Terms
- `"create image"`
- `"gpt image"`
- `"4o image"`
- `"prompt share"`
- `"gpt prompt"` and `"4o prompt"`
- `"create image gpt"`
- `"image gen"`
- `"生成"`
- `"画像"`

---

Figure E.1: Keywords used to query posts, described in Section 3.1. For the initial two weeks following the 4o Image Gen release we favor volume: we query more generic terms, over daily intervals. In later weeks we favor precision: we query more targeted terms often used for sharing image generation results, over weekly intervals. To increase coverage in foreign languages, we also use calligraphic keywords applicable to Chinese and Japanese, while the alphanumeric keywords are sufficient to also cover Romance languages like Spanish and French.

---

**Instructions**

```
You are a helpful assistant that evaluates the relevance of Twitter posts to OpenAI's GPT-4o image-
generation feature. The goal is to assign a relevance score to each tweet.

Scoring Scale
- 1 - Definitely trash
Contains "4o" or "gpt" only by coincidence and has no relation to image generation (e.g., political
commentary, education topics).
- 2 - Very likely irrelevant
Mentions "4o" or "gpt" but clearly not about generating or editing images (e.g., "4o" as slang, or
references to GPT in a purely textual context).
- 3 - Ambiguous
Could plausibly refer to GPT-4o image generation but lacks clear indicators (e.g., "This is insane..." or
 mild excitement without explicit "image" context).
- 4 - Very likely relevant
Contains clear prompt-like language or references to creating or sharing images (e.g., "turn myself into
a cartoon!", "prompt share!", "new prompt").
- 5 - Certainly relevant
Explicitly about using GPT-4o for image generation or editing, often including sample prompts or direct
praise (e.g., "GPT-4o image gen is amazing!", "tried this with GPT-4o image gen, prompt: ...").

Prompt for X data cleaning
- Read the tweet JSON.
- Determine which level best matches the content.
- Output exactly one JSON object with a "score" field set to an integer 1-5.
- If you choose 3, you may optionally add a "note" field (one sentence) explaining the uncertainty.

Input Example
<tweet_json>

Output Example (Score Only)
{"score": 4}
```

---

Figure E.2: Prompt for relevance filtering after raw data collection with GPT-4o (OpenAI, 2024), described in Section 3.1.

```
Instructions

You are an extractor of multimodal prompts for image generation.

You will be given a JSON that represents a Twitter post and its reply tree. Each post in the tree may
contain an image generation prompt; your job is to extract them into unique samples.
For every input, try to extract at least one sample rather than returning an empty list. We want to
extract as many samples as possible, and use a quality score for filtering.
Please output a JSON list of samples in the format ```json [...]```.

## Post to Prompt
Each sample should include the following keys:
{"prompt": <str>, "prompt_modified": <bool>, "post_urls": <list of strs>}

To extract the "prompt":
- Identify each post that discusses a unique image generation task. Set "prompt" as the post text that
describes this task. Be very broad in the definition of "prompt"; any instruction, description, comment,
or question that hints at an image generation task is fine.
- Make a new sample for every new prompt, even if it is a slight modification of another sample's prompt.
- Try to extract the prompt from the post text exactly, without modification. You may modify the prompt
when the modification is obvious, for example, piecing together text from multiple posts or filling in
placeholder text. Set the flag "prompt_modified" to True or False accordingly.
- Omissions of text should not be considered as modifications; you should omit statements that are
obviously not part of the prompt.
- Many main posts say something like "Prompt Below" or "Prompt in Next Comment"; this means that the tree
 is likely to have a really good sample, and the prompt needs to be found in the replies.

To determine "post_urls":
- For each "prompt", set "post_urls" to the urls of posts in the tree that likely contain images that are
 related inputs or outputs for that prompt, which you can determine from the post text.
- Order "post_urls" by importance; the first url should contain the main task information.
- Many replies use a similar prompt as the root post and attach an output image. These should be grouped
in the "post_urls" of the main post. Try to infer if this is happening from the reply text.
- If the reply's text indicates a new task, it should be a new sample. If the reply's text indicates it
is irrelevant to image generation, it should be omitted. If the reply contains no text and an image, it
should be included in "post_urls" so that it can be further processed later.
- Each url/post should appear at most once; images should not be duplicated across samples.

## Quality Score
Each sample should also classify the prompt quality:
{"quality": <str>}

To classify "quality":
- Classify the quality as one of the following categories: ["Benchmark", "Analysis", "Trash"].
- "Benchmark" are the highest quality prompts, which instruct a single coherent image generation task,
that can be used for benchmarking. Be fairly strict about the quality.
- "Analysis" are moderate quality prompts that are not in "prompt" format, which are often comments or
questions relevant to image generators but do not query a specific task, and are still usable for
analysis.
- "Trash" are low quality prompts that have no clear task or are clearly irrelevant. Our focus is on
OpenAI's gpt-image-1 or 4o image generation; if the post clearly uses another model or platform like DALL-
E, Stable Diffusion, some video generator, etc. it should be classified as "Trash".
- Make sure to collect as many "Analysis" samples as possible, while maintaining relevancy. For these
samples, set "prompt" to be the relevant text or commentary about image generation.

## Community Feedback
Each sample should contain a list of community feedback:
{"community_feedback": [{"post_url": <str>, "feedback": <str>}, ...]}

To extract "community_feedback":
- For each post in the tree, determine whether it discusses the sample's success / quality (e.g., "really
 cool", "does not work", etc.).
- If a post obviously does not have feedback, do not include it.
- The feedback may come from the main post's author or from other authors in the replies.
- Include the full feedback text without modification such that there is sufficient context, but also
omit obviously irrelevant text.
- Each url/post should appear at most once in the "community_feedback"; feedback should not be duplicated
 across samples.

json_post_tree: <tree>
extracted:
```

Figure E.3: Prompt for tree-to-sample extraction with GPT-4o (OpenAI, 2024), described in Section 3.2.

```
Instructions

You are an extractor of multimodal prompts for image generation.

Your job is to process input-output image pairs from raw user prompts for image generation collected from
 Twitter.
You will be given a prompt, a dictionary mapping image ids to images, and a dictionary mapping image ids
to post urls.
Please output a JSON list of samples in the format ```json [...]```.

## Image Classification
Each sample should include the following keys, which categorize images as inputs or outputs:
{"inputs": <list of ids>, "outputs": <list of ids>, "post_urls": <list of strs>}

To classify "inputs" and "outputs":
- Inputs, combined with the prompt, should produce a fully specified and coherent image generation task.
- Outputs should be plausible results given the inputs and prompt.
- You may encounter tasks like text-to-image generation (no inputs), image editing (one input), or multi-
image conditioned generation (multiple inputs).
- Set "post_urls" the list of urls associated with the inputs and outputs. Order "post_urls" by
importance; the first url should contain the main task information.

General rules:
- Each category is mutually exclusive. Each image should not be assigned to multiple categories.
- Some images are low quality and irrelevant to any task. They should not be assigned to any category.
- Some samples are low quality where it is not possible to extract any coherent task. Simply return an
empty dictionary {}.
- If there are no relevant images, assign an empty list [] to the category.
- If an image is duplicated, use the smaller index as the id and ignore the others.
- Each id should appear at most once. Each post_url should appear at most once.

## Fill in the Blank
The input prompt may be a "fill in the blank" case with placeholders. Infer these placeholders and update
 the following keys:
{"prompt": <str>, "prompt_fill_blank": <bool>}

To update the prompt if it is "fill in the blank":
- If the prompt is not "fill in the blank", which should happy the majority of the time, you should by
default copy the input prompt exactly and set "prompt_fill_blank" to False.
- Otherwise update the prompt and update the flag "prompt_fill_blank" to True.
- Often fill in the blank prompts include brackets of the form "[keyword]".
- Often you can infer the right keyword to replace the placeholder using the output images.
- Often you will generate multiple infilled prompts, because there are often multiple output images that
represent different instantiations with different sets of keywords.
- Only fill in the blank only when it makes sense to do so, and when you are fairly confident about what
the keyword should be. Otherwise, if highly uncertain, don't "fill in the blank".
- You should make a new sample for each new instantiation of the "fill in the blank". If there are
multiple outputs that infill with different keywords, you should create multiple samples.

## Screenshots of Conversations
For special images that show a screenshot of a conversation with the image generator, mark their image id:

{"conversation": <id>}

To extract a "conversation":
- For each conversation, you should create a new sample that represents the task expressed in the
conversation.
- If there exist multiple images showing screenshots of the same conversation, select the main one
showing the most task information and omit the others.
- Combined related samples and their fields like "inputs", "outputs", "post_urls", "prompt" to minimize
redundancy.
- A conversation is defined as a screenshot that shows a conversation (which may involve a prompt and
image(s)) in OpenAI's ChatGPT window.
- If the image shows any other platform, it is not a conversation.
- If the image generation task is not clear (e.g., the screenshot seems to be using ChatGPT's LLM rather
than image generation capabilities, the screenshot is extremely low quality, the images are extremely
small), it is also not a conversation.
- If the sample does not contain a conversation, set "conversation" to the empty string "".

prompt: <prompt>
images: <images>
images_to_posts: <images_to_posts>
extracted:
```

Figure E.4: Prompt for multimodal processing with GPT-4o (OpenAI, 2024), described in Section 3.3.

```
Instructions

You are an extractor of multimodal prompts for image generation.

Your job is to extract the text prompt and bounding boxes of individual images from screenshots of
conversations with an image generator.
You will also be provided relevant text that may be helpful for determining the input and output images
from the screenshot.

Please output only a valid JSON dictionary according to this schema:
```json {"prompt": <str>, "inputs": <list of bounding boxes>, "outputs": <list of bounding boxes>}```

To extract "prompt":
- If there is text, run OCR and extract the raw text input by the user exactly.
- The extracted text should produce a fully specified and coherent image generation task; ignore other
irrelevant text.
- If there is no relevant text, output the empty string "".

To extract "inputs" and "outputs":
- Extract a list of bounding boxes for every individual image.
- Each bounding box should be formatted as [x1, y1, x2, y2]; (x1, y1) is the top-left and (x2, y2) is the
 bottom-right.
- Also sort bounding boxes as "inputs" vs. "outputs" of the image generator.
- The extracted images should produce a fully specified and coherent image generation task; ignore other
irrelevant images.
- Each image should only appear once. Ignore exact duplicates.
- If there are no "inputs" output an empty list [].
- If there are no "outputs" output an empty list [].

relevant_text: <relevant_text>
images: <images>
extracted:
```

Figure E.5: Prompt for parsing screenshots of conversations with Qwen-2.5-VL (Bai et al., 2025), described in Section 3.3.

# F  AUTOMATIC EVALUATION METRICS

**Overall Metrics.** In Figure F.1 we depict the prompt used for VLM-as-a-judge in our overall benchmark evaluation. We follow the "single answer grading setup" of MT-Bench (Zheng et al., 2023), and convert absolute scores into pseudo pairwise comparisons, which can be used to compute the win rate.

**Specialized Metrics.** We display our prompt for rating the accuracy of rendered text in Figure F.2, and classifying the applicability of each sample to each specialized metric in Figure F.3.

```
Instructions

Please act as an impartial judge and evaluate the quality of the image output produced by an image
generation model in response to an input instruction (expressed via text and/or image(s)).

Begin your evaluation by forming your own expectation of what a good output should look like for the
given prompt. Describe this briefly before judging the output.

Then compare the model's output with your expectation. Point out errors, inaccuracies, or failures to
follow the instruction, and identify missing details that would make the output better satisfy the
instruction.

Make sure to consider the following factors equally:
- **Prompt Following**: Does the output interpret the text correctly and execute the requested task
accurately?
- **Reference Fidelity**: Does the output preserve key details from the input images when relevant?
- **Realism and Aesthetics**: Is the output photorealistic (e.g., accurate anatomy, no artifacts, etc.)
and visually appealing (e.g., balanced colors, well-framed composition, etc.) when relevant?

After providing your explanation, please rate the output on a scale of 1 to 10 by strictly following this
 format: "[[rating]]", for example: "Rating: [[5]]".

<|The Start of Input Instruction|>
input_prompt: <input_prompt>
input_images: <input_images>
<|The End of Input Instruction|>

<|The Start of Model Output|>
output_image: <output_image>
<|The End of Model Output|>
```

Figure F.1: Prompt for automatic evaluation with GPT-4o (OpenAI, 2024), Gemini 2.0 (Team et al., 2023), and Qwen2.5-VL-32B-Instruct (Bai et al., 2025), described in Section 5. Our prompt closely follows the format from MT-Bench (Zheng et al., 2023), but adapted for rating image generation outputs.

```
Instructions

Check if all text in the image is accurate and readable.
For exact copy requests: spelling, punctuation, grammar match exactly, with no missing or extra
characters, and text is not cropped.
For created text: content is coherent, relevant, and fits the available space and design.
Begin your evaluation by reading through the image and OCR the text.
Point out spelling errors, punctuation errors, grammar errors, and missing characters of the text.
Point out if the text is cropped.
Then, look at the image again and check if the text is coherent, relevant, and fits the available space
and design.
Please rate the output on a scale of 1 to 10 by strictly following this format: "[[rating]]", for example:
 "Rating: [[5]]".

<|The Start of Input Instruction|>
input_prompt: <input_prompt>
<|The End of Input Instruction|>

<|The Start of Model Output|>
output_image: <output_image>
<|The End of Model Output|>
```

Figure F.2: Prompt for judging text rendering accuracy with GPT-4o (OpenAI, 2024), described in Section 5.2.

```
Instructions

   For each metric in the provided list, decide if it is applicable for the given image generation
instruction.
   The instruction can be defined with text and/or images. Some instructions may contain no input images.
   If is a metric is marked as applicable, it will be used as an axis to score and rank outputs for the
given input.

   ## Metric List
   <metric_list>

   ## Output Format
   Respond only with a JSON dictionary containing all the metric names as keys, and the value 0 (is not
applicable) or 1 (is applicable).
   Also include a short global rationale for your overall decision-making process.
   ```json
   {
     "<metric1>": <integer 0 or 1>,
     "<metric2>": <integer 0 or 1>,
     ...
     "rationale": "<a short rationale, 20 words or less>"
     "prompt": "<the input prompt repeated again>"
   }
   ```

   ## Your Turn
   task: <task>
   input_prompt: <input_prompt>
   input_images: <input_images>
   metrics:
```

The <metric_list> is replaced with the defined metrics name, its description, and its applicability:

```
Name: "Face Identity Preservation"
Description:
  Check if the person's identity matches the reference or intended person, keeping facial structure and
distinctive features the same.
  Examples to Penalize: Changes in hairstyle, beard length, scars, facial expression, accessories, etc.
that do not match the reference.
Applicability:
  This metric is often applicable, especially for tasks involving subject-driven generation.
  However, it is not applicable when:
  - The prompt does not explicitly or implicitly request face identity preservation.
  - No person's face is visible (because there are no people, or faces are occluded).
  - The task is stylization, where the creative freedom allows for many valid outputs and correctness is
too subjective.

Name: "No Color Shift"
Description:
  Check if the overall color tone, brightness, and contrast match the reference or intended look.
  Examples to Penalize: Added yellow tint, overexposure, or darkening compared to the reference.
Applicability:
  This metric is often applicable, especially for tasks like local editing.
  However, it is not applicable when:
  - The task is colorization or image-to-image translation, where color change is inherent to the task.

Name: "Spatial Position Preservation"
Description:
  Check if the structure and spatial layout of the reference are copied correctly, including positions,
relative layout, and scale of key objects.
  Examples to Penalize: A dog is slightly moved from its original position during stylization; a table
that was centered is shifted.
Applicability:
  This metric is only applicable for tasks that involving image-to-image translation, stylization, or
local editing that requires strict structure preservation.
  However, it is not applicable when:
  - The prompt does not expect the resulting image to be strictly preserving spatial structure with the
reference image.
  - The prompt can allow some structure changes (eg, sketch-to-image, 2D-to-3D stylization)

Name: "Text Rendering Accuracy"
Description:
  Check if rendered text contains mistakes that hinder readability.
  Examples to Penalize: Characters are garbled; there are missing or extra characters; there is incorrect
 spelling or punctuation; there is incorrect grammar.
Applicability:
  This metric is often applicable, but only when the prompt explicitly requests rendered text.
```

Figure F.3: Prompt for classifying the set of applicable samples for each specialized metric with GPT-4o (OpenAI, 2024), described in Section 5.2. The number of applicable samples is as follows: Face Identity Preservation (244), No Color Shift (271), Spatial Position Preservation (180), Text Rendering (240).

## G LLM DISCLOSURE

Some portions of this work were generated with the assistance of large language models (LLMs). Their primary role was to support editing, rephrasing, and formatting of existing text to improve clarity and readability. While human authors created and reviewed the core content, LLMs were used as a tool to streamline refinement and presentation. All factual information, analysis, and conclusions remain the responsibility of the authors, and every effort has been made to ensure accuracy and integrity.

