# OpenReview forum: "Constantly Improving Image Models Need Constantly Improving Benchmarks"
_ICLR.cc/2026/Conference — ICLR 2026 Poster_

### Official Review · Reviewer_eprr · 2025-10-31

**Soundness:** 3
**Presentation:** 3
**Contribution:** 4
**Rating:** 6
**Confidence:** 4

**Summary:**

This paper addresses the growing gap between the rapid progress of image generation models and the static nature of existing benchmarks. It introduces ECHO, where instead of relying on predefined and manually curated tasks, the framework builds its challenges by mining social media posts about new image models (the case study for this paper is GPT-4o Image Gen) to construct an in-the-wild, re-runnable benchmark. It then details an LLM and VLM-based processing pipeline, where the authors extract structured <input, output, feedback> tuples from messy social media threads and curate a benchmark of around 35K samples. The paper demonstrates that ECHO captures creative, complex, and evolving tasks not present in existing benchmarks, provides stronger differentiation among models, and motivates the development of new quantitative metrics (e.g., colour shift, text rendering accuracy, identity preservation) based on community feedback.

**Strengths:**

1. The paper's central premise (benchmarks must dynamically evolve with model capabilities and user behaviours) is a highly significant and timely contribution. The idea of using community-generated evidence from social media as a source is a novel solution.
2. The paper is clearly written, visually rich, and easy to follow. The proposed pipeline (ECHO) is detailed and well-motivated. The analysis demonstrates how ECHO surfaces novel tasks, uncovers model failure modes, and differentiates state-of-the-art models.
3. It is an important contribution to how we evaluate image generation models. The framework could inspire new adaptive benchmarking protocols in other modalities.
4. Traditional benchmarks for image generation often rely on abstract metrics that correlate poorly with human preference. ECHO’s feedback-derived metrics aim to capture human-relevant error dimensions.

**Weaknesses:**

1. Although ECHO collects a large and diverse dataset, the quality control process heavily depends on LLM and VLM filtering. The paper does not provide any quantitative validation on how accurate the pipeline is. How do the authors handle bias or noisy samples in their final benchmark? The fill-in-the-blank prompts (Section 3.3) could also introduce hallucinated prompts.
2. The authors analyze bigrams to assess the linguistic diversity of the benchmark, but this is a coarse view of the benchmark. A more informative approach would be to use an LLM to cluster each datapoint into broader task categories, which will be more meaningful to understand the benchmark distribution. It would also reveal potential imbalances, there could be an overrepresentation of certain tasks, which is not currently captured in the analysis.

**Questions:**

1. The framework currently relies on (presumably) Western-centric, English-language social media. How feasible or what challenges would arise to extend ECHO to non-English or region-specific platforms and mitigate this sampling bias.
2. Can the authors provide a more granular breakdown of the benchmark's composition, as suggested in W2. An analysis of task clusters would be far more insightful than bigrams and would help identify which specific capabilities ECHO is truly testing and how balanced it is.
3. Based on this analysis, which tasks in the benchmark are most difficult (i.e., highest failure rate for all models) and which are comparatively easier? This would be invaluable for guiding future research.

---

> ### Author Response · Authors · 2025-11-26
> **Response to Reviewer eprr**
>
> Thank you for your suggestions! Below, we validate the pipeline’s accuracy, explore extending ECHO to non-English platforms, analyze task clusters, and report the least and most difficult tasks.
>
> ---
>
> 1. *[Q]uantitative validation on how accurate the [LLM and VLM filtering] pipeline is.*
>
> We design ECHO such that it is decomposed into clear-cut and modular sub-steps, and empirically validate the automated pipeline’s accuracy in Table 1 below. To quantify accuracy, we take a subset of samples and hand label the ground-truth for each sub-step. For “Classify Input-Output” we take the unstructured image pool then label each as an input or output, for “Fill-in-the-Blank” we take the template and output image then annotate short keyword completions, for “Quality Filter” we take the sample then label each as Benchmark, Analysis, or Discard (corresponding to high, moderate, and low quality). We also add this analysis to Sec. A.5 of the Appendix.
>
> **As seen in Table 1, each automated sub-step achieves at least 80\% accuracy.** Furthermore, for the final benchmark evaluated in Sec. 5, we perform an additional manual validation step where we discard any incorrectly processed samples. This removes the possibility of noisy samples or hallucinated prompts in the final benchmark.
>
> **Table 1. ECHO pipeline accuracy, computed against 100 human-labeled examples.**
> | | Description | Accuracy |
> |-|-|-|
> | Classify Input-Output | % correct classification in {Input, Output} | 0.9200 |
> | Fill-in-the-Blank | % ground-truth keywords matched | 0.8382 |
> | Quality Filter | % correct classification in {Benchmark, Analysis, Discard} | 0.8000 |
>
> ---
>
> 2. *The framework currently relies on (presumably) Western-centric, English-language social media.*
>
> To clarify, we also make an effort to collect the non-English data that exists on Twitter/X; we design our keyword searches such that they can capture posts in Romance languages like Spanish and French as well as calligraphic languages like Chinese and Japanese (see Fig. E.1 of the Appendix).
>
> **We also show that it is straightforward to apply the same ECHO pipeline to Douyin, a non-English and short-form video platform that is very different from Twitter/X.** Many Douyin users post screenshots of their conversations with 4o Image Gen, which the “Conversation Screenshot” step of our pipeline is able to automatically extract.
>
> In a small beta test, we collected 100 Douyin posts and postprocessed them with the ECHO pipeline, with the following quality classifications: {Benchmark: 18, Analysis: 41, Discard: 41}. We also show examples in Fig. A.3 of the Appendix.
>
> ---
>
> 3. *An analysis of task clusters.*
>
> **We add a more granular breakdown of task clusters, comparing both ECHO and prior datasets, to Sec. A.1 of the Appendix.** To obtain the clusters, we apply the topic modeling pipeline from Arena Explorer [1], which embeds prompts with OpenAI’s text-embedding-3-large, reduces the embedding dimensionality with UMAP, then clusters with HDBSCAN. Each cluster is then labeled with GPT4o based on its associated prompts.
>
> Compared with prior image-to-image datasets limited to object and scene editing, ECHO surfaces new tasks like “YouTube Thumbnail Design” and “3D Rendering and Retexturing” (see Fig. A.1). While prior text-to-image datasets focus on generating objects or fantastical imagery, ECHO also tests real-world applications like “Educational Visual Storytelling” and "Food and Product Photography" (see Fig. A.2).
>
> [1] Tang et. al. Arena Explorer: A Topic Modeling Pipeline for LLM Evals & Analytics. https://news.lmarena.ai/arena-explorer.

---

> ### Author Response · Authors · 2025-11-26
> **Response to Reviewer eprr**
>
> 4. *[W]hich tasks in the benchmark are most difficult [...] and which are comparatively easier*
>
> **We show the performance of different models on individual tasks in Tables 5 and 6 below.** Based on the representative tasks shown in Figures B.3-B.6 of the Appendix, we curate a task list then use a VLM to label each prompt with a task. We opt to use these top-down tasks rather than the bottom-up clusters from response #3 above, as they are more fine-grained and better aligned with established tasks of interest to the research community.
>
> We then rank task difficulty based on the win rate of the best-performing model on that individual task (a lower rank means the task is harder). For the image-to-image split, we observe that the hardest task is “Template-Based Product Generation” and the easiest is  “Virtual Try-On.”  For the text-to-image split, we observe that the hardest task is “Generic Text-to-Image” and the easiest is “Info Graphics.”
>
> **Table 5. Model win rate stratified into individual tasks, image-to-image split.** For each task, we bold the best-performing model.
> |Task|Task Difficulty Rank (1=hardest)|4o Image Gen|Nano Banana|Gemini 2.0 Flash|Bagel Think|Bagel|Flux Kontext|LLM+Diffusion|Anole|
> |:-|:-|:-|:-|:-|:-|:-|:-|:-|:-|
> |Template-Based Product Generation|1|**0.78**|0.76|0.58|0.46|0.41|0.45|0.17|0.11|
> |Novel View Synthesis|2|**0.81**|0.60|0.51|0.48|0.49|0.42|0.27|0.16|
> |Colorization|3|**0.81**|0.75|0.57|0.48|0.41|0.48|0.12|0.02|
> |Multi-Image Subject-Driven Generation|4|**0.81**|0.76|0.55|0.44|0.45|0.33|0.21|0.03|
> |Code-Based Style Transfer|5|**0.82**|0.48|0.47|0.57|0.59|0.51|0.13|0.07|
> |Text Enhancement|6|**0.82**|0.65|0.57|0.43|0.45|0.45|0.22|0.04|
> |Ad Creation|7|**0.82**|0.81|0.53|0.46|0.37|0.41|0.23|0.08|
> |Real-to-Sim|8|**0.82**|0.77|0.41|0.43|0.47|0.48|0.11|0.00|
> |Sim-to-Real|9|**0.83**|0.61|0.57|0.43|0.47|0.52|0.19|0.07|
> |Virtual Try-On|10|0.76|**0.83**|0.57|0.39|0.38|0.57|0.10|0.00|
>
> **Table 6. Model win rate stratified into individual tasks, text-to-image split.** For each task, we bold the best-performing model.
> |Task|Task Difficulty Rank (1=hardest)|4o Image Gen|Nano Banana|LLM+Diffusion|Gemini 2.0 Flash|Bagel Think|Bagel|Flux Kontext|Anole|
> |:-|:-|:-|:-|:-|:-|:-|:-|:-|:-|
> |Generic Text-to-Image|1|**0.73**|0.72|0.60|0.54|0.49|0.43|0.39|0.10|
> |Logo Creation|2|**0.80**|0.74|0.54|0.59|0.43|0.38|0.46|0.07|
> |Comic Creation|3|**0.82**|0.78|0.61|0.50|0.45|0.42|0.36|0.07|
> |Poster Creation|4|**0.84**|0.80|0.57|0.58|0.38|0.36|0.36|0.11|
> |Info Graphics|5|**0.87**|0.79|0.71|0.52|0.40|0.31|0.35|0.09|

---

### Official Review · Reviewer_dbsG · 2025-11-01

**Soundness:** 3
**Presentation:** 3
**Contribution:** 3
**Rating:** 4
**Confidence:** 4

**Summary:**

The paper introduces a framework ECHO, that constructs benchmarks by directly probing social media posts from platforms like twitter. The framework is applied to GPT 4o Image Gen, and thereby a collection of 35000 prompts are curated, which are those that are directly used by real-life users. Resultantly, the benchmarks consists of complex and realistic tasks, not included in previous benchmarks. By applying such prompts, state-of-the-art generative models are re-evaluated, and the discovered indicators from the posts help evaluate the models further.

**Strengths:**

1. The benchmark is unique, and is of utmost importance, as it ties the general public opinion with model performance.
2. The authors have put in substantial efforts to extract the prompts from the complex tweet chains.
3. The framework uncovers significant model failures observed by the users, as shown in Fig. 4.

**Weaknesses:**

1. While I agree that the large-scale images generated from the prompts cannot be manually evaluated, the VLM-human correlation seems quite low, raising questions on the evaluations.
2. It is unclear how many images have been generated by each generative model for evaluation.
3. The benchmark consists of several tasks, while Fig. 6 summarizes them. It would be good to capture how the different models on the individual tasks, as virtual try on, novel view synthesis are themselves quite significant as tasks.
4. Fidelity, Faithfulness and Diversity - the three most important metrics related to text-to-image generations should have been discussed in more detail.
5. [minor] The resulting observations are similar to the expected results - the closed-source models generally outperform the open-sourced ones. However, that being said, the curated prompts seem useful, as they are obtained from real-life users.

**Questions:**

1. How many images per model did the authors evaluate on the constructed benchmark?
2. Did the authors try the traditional fidelity and faithfulness metrics? Wherever applicable, how diverse are the generations by the different models? This question is important as the prompts are unique in the proposed benchmark.
3. Counting and hallucinations have been raised by users as failure cases, and these have been well-established problems in literature. Did authors try measuring them, especially for newer models like FLUX Kontext, GPT etc?

---

> ### Author Response · Authors · 2025-11-26
> **Response to Reviewer dbsG**
>
> Thank you for your thoughtful feedback! Below, we compare the correlation of other evaluators with human judgements (#2), clarify the number of images evaluated per model (#3), report model performance on individual tasks (#4), evaluate the suggested fidelity, faithfulness, and diversity metrics (#1), and investigate special cases like counting and hallucination (#5).
>
> ---
>
> 1. *Fidelity, Faithfulness and Diversity - the three most important metrics related to text-to-image generations should have been discussed in more detail.*
>
> We acknowledge that it would be useful to the reader to understand the Fidelity, Faithfulness and Diversity of each of these models. Therefore, we report the FID score to evaluate Fidelity, the CLIP Text-Image score to evaluate Faithfulness, and the average DreamSim score per sample (following [1]) to evaluate diversity.
>
> For FID and DreamSim, we generate 10 images per prompt for each model, resulting in 7,770 images on the Image-to-Image split and 10,000 images on the Text-to-Image split. Due to compute constraints, we report FID and DreamSim for the following models: Bagel, Flux Kontext, Nano Banana, LLM+Diffusion, and 4o Image Gen. CLIP text-image scores are reported for all models. The results are summarized below.
>
> **Table 1. Fidelity (FID Score).** Lower scores mean better visual fidelity. For the reference set, we use 10k images from the COCO 2017 validation set.
> |Model|Text-to-Image|Image-to-Image|
> |-|-|-|
> |4o Image Gen|85.35|83.79|
> |Nano Banana|76.96|70.64|
> |LLM+Diffusion|84.86|77.41|
> |Flux Kontext|87.16|86.04|
> |Bagel|76.21|85.88|
>
>
> **Table 2. Faithfulness (CLIP Score).** Higher scores mean better text-image faithfulness.
> |Model|Text-to-Image|Image-to-Image|
> |-|-|-|
> |4o Image Gen|0.3672|0.2703|
> |Gemini|0.3555|0.2561|
> |Nano Banana|0.3574|0.2461|
> |LLM+Diffusion|0.3468|0.1675|
> |Flux Kontext|0.3287|0.2604|
> |Bagel-Think|0.3281|0.2636|
> |Bagel|0.3218|0.2717|
> |Anole|0.2598|0.2018|
>
> **Table 3. Diversity (DreamSim).** Higher scores mean better diversity.
> |Model|Text-to-Image|Image-to-Image|
> |-|-|-|
> |4o Image Gen|0.1946|0.1139|
> |Nano Banana|0.2912|0.1568|
> |LLM+Diffusion|0.3479|0.3096|
> |Flux Kontext|0.2827|0.1174|
> |Bagel|0.3484|0.1920|
>
>
> [1] Gandikota et. al. Distilling Diversity and Control in Diffusion Models. arXiv 2025.
>
> ---
>
> 2. *While I agree that the large-scale images generated from the prompts cannot be manually evaluated, the VLM-human correlation seems quite low*
>
> We would like to underscore that evaluating the quality of image generation remains a longstanding open problem, and it is not only very difficult but also out of scope for our work. We mainly follow the practice adopted in recent literature (e.g., Bagel), which also uses a VLM-as-a-judge to score generated images.
>
> To address the concern about low VLM-human correlation, we additionally conducted a human-correlation study under the same setting using two widely used automatic metrics: CLIP text-image similarity and DreamSim image-image similarity.
>
> **Table 4. CLIP score:**
> ||τ|p|
> |-|-|-|
> |Image-to-Image|0.0936|0.0089|
> |Text-to-Image|0.0751|0.0221|
>
> **Table 5. DreamSim score:**
> ||τ|p|
> |-|-|-|
> |Image-to-Image|–0.1222|0.986|
> |Text-to-Image|–0.0315|0.746|
>
> As shown, these automatic scores have similarly weak or even worse correlation with human preferences compared to VLM-as-a-judge. In other words, replacing the VLM judge with CLIP or DreamSim does not solve the problem; all current automatic metrics struggle to directly reflect fine-grained human preference.

---

> ### Author Response · Authors · 2025-11-26
> **Response to Reviewer dbsG**
>
> 3. *It is unclear how many images have been generated by each generative model for evaluation.*
>
> We originally evaluated 1,777 samples in total for each model, with one image generated per sample. We also perform an ablation study on the number of images generated per sample, reported in Tables 6 and 7 below. Specifically, we evaluated more samples by generating 10 images per sample per model, and did not see a significantly different result in terms of model ranking of Figure 6. The results also stabilize as the number of rounds increases.
>
> **Table 6. Text-to-Image winrate when varying number of outputs per sample.** Winrates are computed from 1k-10k total images (for O=1 to O=10).
> |Model|O=1|O=2|O=3|O=4|O=5|O=6|O=7|O=8|O=9|O=10|
> |:-|:-:|:-:|:-:|:-:|:-:|:-:|:-:|:-:|:-:|:-:|
> |4o Image Gen|0.70|0.71|0.71|0.72|0.72|0.72|0.72|0.72|0.72|0.72|
> |Nano Banana|0.69|0.70|0.70|0.70|0.70|0.70|0.70|0.70|0.71|0.71|
> |LLM+Diffusion|0.52|0.52|0.52|0.52|0.52|0.52|0.52|0.52|0.52|0.52|
> |Bagel|0.31|0.30|0.29|0.28|0.28|0.28|0.28|0.28|0.28|0.28|
> |Flux Kontext|0.28|0.28|0.28|0.29|0.29|0.29|0.29|0.29|0.29|0.29|
>
> **Table 7. Image-to-Image winrate when varying number of outputs per sample.** Winrates are computed from 777-7.77k total images (for O=1 to O=10).
> |Model|O=1|O=2|O=3|O=4|O=5|O=6|O=7|O=8|O=9|O=10|
> |:-|:-:|:-:|:-:|:-:|:-:|:-:|:-:|:-:|:-:|:-:|
> |4o Image Gen|0.82|0.81|0.81|0.81|0.81|0.82|0.82|0.81|0.81|0.81|
> |Nano Banana|0.64|0.63|0.62|0.62|0.62|0.62|0.62|0.62|0.62|0.62|
> |Bagel|0.45|0.46|0.46|0.46|0.46|0.46|0.46|0.46|0.46|0.46|
> |Flux Kontext|0.42|0.43|0.43|0.43|0.43|0.43|0.43|0.43|0.43|0.43|
> |LLM+Diffusion|0.17|0.18|0.18|0.18|0.18|0.18|0.18|0.18|0.18|0.18|
>
> ---
>
> 4. *It would be good to capture how the different models on the individual tasks, as virtual try on, novel view synthesis are themselves quite significant as tasks.*
>
> We show the performance of different models on individual tasks in Tables 8 and 9 below. Based on the representative tasks shown in Figures B.3-B.6 of the Appendix, we curate a task list then use a VLM to label each prompt with a task.
>
> We then rank task difficulty based on the win rate of the best-performing model on that individual task (a lower rank means the task is harder). For the image-to-image split, we observe that the hardest task is “Template-Based Product Generation” and the easiest is  “Virtual Try-On.”  For the text-to-image split, we observe that the hardest task is “Generic Text-to-Image” and the easiest is “Info Graphics.”
>
> **Table 8. Model win rate stratified into individual tasks, image-to-image split.** For each task, we bold the best-performing model.
> |Task|Task Difficulty Rank (1=hardest)|4o Image Gen|Nano Banana|Gemini 2.0 Flash|Bagel Think|Bagel|Flux Kontext|LLM+Diffusion|Anole|
> |:-|:-|:-|:-|:-|:-|:-|:-|:-|:-|
> |Template-Based Product Generation|1|**0.78**|0.76|0.58|0.46|0.41|0.45|0.17|0.11|
> |Novel View Synthesis|2|**0.81**|0.60|0.51|0.48|0.49|0.42|0.27|0.16|
> |Colorization|3|**0.81**|0.75|0.57|0.48|0.41|0.48|0.12|0.02|
> |Multi-Image Subject-Driven Generation|4|**0.81**|0.76|0.55|0.44|0.45|0.33|0.21|0.03|
> |Code-Based Style Transfer|5|**0.82**|0.48|0.47|0.57|0.59|0.51|0.13|0.07|
> |Text Enhancement|6|**0.82**|0.65|0.57|0.43|0.45|0.45|0.22|0.04|
> |Ad Creation|7|**0.82**|0.81|0.53|0.46|0.37|0.41|0.23|0.08|
> |Real-to-Sim|8|**0.82**|0.77|0.41|0.43|0.47|0.48|0.11|0.00|
> |Sim-to-Real|9|**0.83**|0.61|0.57|0.43|0.47|0.52|0.19|0.07|
> |Virtual Try-On|10|0.76|**0.83**|0.57|0.39|0.38|0.57|0.10|0.00|
>
> **Table 9. Model win rate stratified into individual tasks, text-to-image split.** For each task, we bold the best-performing model.
> |Task|Task Difficulty Rank (1=hardest)|4o Image Gen|Nano Banana|LLM+Diffusion|Gemini 2.0 Flash|Bagel Think|Bagel|Flux Kontext|Anole|
> |:-|:-|:-|:-|:-|:-|:-|:-|:-|:-|
> |Generic Text-to-Image|1|**0.73**|0.72|0.60|0.54|0.49|0.43|0.39|0.10|
> |Logo Creation|2|**0.80**|0.74|0.54|0.59|0.43|0.38|0.46|0.07|
> |Comic Creation|3|**0.82**|0.78|0.61|0.50|0.45|0.42|0.36|0.07|
> |Poster Creation|4|**0.84**|0.80|0.57|0.58|0.38|0.36|0.36|0.11|
> |Info Graphics|5|**0.87**|0.79|0.71|0.52|0.40|0.31|0.35|0.09|

---

> ### Author Response · Authors · 2025-11-26
> **Response to Reviewer dbsG**
>
> 5. *Did authors try measuring [counting and hallucinations], especially for newer models like FLUX Kontext, GPT etc?*
>
> We agree that Counting and Hallucination would be an interesting extension of the specialized metrics discussed in Sec. 5.2. We analyze these cases in Tables 10 and 11 below, as scored by VLM-as-a-judge (see prompt in Appendix Fig. A.4). The VLM is asked to score whether the generated image exhibits counting errors or hallucination on a scale of 1-10 (where 1 = contains significant errors and 10 = perfect and without error).
>
> These results show a consistent trend: 4o Image Gen and Nano Banana perform the strongest with minimal errors in counting or hallucination, while Flux Kontext and Bagel lag behind. The Image-to-Image setting is generally more challenging, especially for LLM+Diffusion and Anole.
>
> **Table 10. Counting:**
> |Model|Text-to-Image|Image-to-Image|
> |-|-|-|
> |4o Image Gen|9.80|9.78|
> |Gemini 2.0 Flash|9.21|8.15|
> |Nano Banana|9.65|8.81|
> |LLM+Diffusion|9.21|3.26|
> |Flux Kontext|8.23|7.40|
> |Bagel-Think|8.50|8.13|
> |Bagel|8.33|7.88|
> |Anole|5.04|1.88|
>
> **Table 11. Hallucination:**
> |Model|Text-to-Image|Image-to-Image|
> |-|-|-|
> |4o Image Gen|9.58|9.55|
> |Gemini 2.0 Flash|8.55|7.28|
> |Nano Banana|9.17|8.47|
> |LLM+Diffusion|8.23|2.56|
> |Flux Kontext|6.90|6.78|
> |Bagel-Think|7.47|7.42|
> |Bagel|7.18|7.00|
> |Anole|4.02|1.91|

---

### Official Review · Reviewer_Fp5T · 2025-11-03

**Soundness:** 3
**Presentation:** 3
**Contribution:** 3
**Rating:** 6
**Confidence:** 4

**Summary:**

The paper introduces ECHO (Extracting Community Hatched Observations), a framework for constructing adaptive, data-driven benchmarks for image generation models based on real-world user interactions on social media. Motivated by the observation that existing evaluation datasets lag behind rapidly evolving generative capabilities, the authors propose leveraging posts from platforms such as Twitter/X to capture emergent tasks, user prompts, and qualitative feedback surrounding newly released models (e.g., GPT-4o Image Gen). ECHO systematically extracts multimodal data, comprising textual prompts, reference images, and community commentary, using a pipeline that integrates large language and vision–language models (LLMs/VLMs) to filter, contextualize, and structure these inputs into benchmark-ready samples.

Empirically, the authors curate a dataset of over 35,000 social media posts and develop a benchmark subset containing approximately 1,700 text-to-image and image-to-image tasks. Using this dataset, they evaluate eight leading generative models, showing that ECHO better distinguishes performance differences than conventional benchmarks such as GEdit or CompBench. Beyond quantitative evaluation, the framework also transforms recurring user observations (e.g., color shift, identity drift, text rendering errors) into measurable diagnostic metrics, providing a mechanism to align benchmarking with authentic user concerns. The paper concludes that such community-grounded, continuously updatable benchmarks can serve as a scalable and dynamic alternative to static evaluation paradigms for assessing progress in generative modeling.

**Strengths:**

1. The paper articulates a timely problem: benchmarks for generative models cannot keep pace with emerging user behavior. The introduction (pp. 1–2) effectively grounds this issue using the example of “Ghiblification” — a community-invented use case of GPT-4o that no prior benchmark captured. This framing convincingly motivates ECHO.

2. The ECHO pipeline (Figure 2) is the paper’s technical core: Collect relevant social posts, Reconstruct context across replies, Process multimodal data (text + images + screenshots), Filter and classify for quality and benchmarking. The pipeline is modular, well-illustrated, and generalizable. The multimodal LLM+VLM processing steps (using GPT-4o and Qwen-2.5-VL) are particularly novel — they turn noisy social media content into structured benchmark samples.

3 Empirical Validation: The authors benchmark 8 models (open-source and proprietary) across the new dataset.

4 Turning user complaints into quantitative metrics (color shift, face identity, text rendering) is an interesting idea. Figure 8 (page 9) demonstrates these metrics’ power to confirm qualitative observations (e.g., GPT-4o’s “yellow tint” or identity drift). This “closing the loop” contribution elegantly connects community perception and model evaluation.

**Weaknesses:**

1. ECHO’s exclusive reliance on Twitter/X introduces substantial platform- and demographic-specific biases. As acknowledged in *Appendix A*, trending phenomena such as the “Ghibli style” disproportionately influence the sample composition, leading to a skewed task distribution that may compromise benchmark representativeness and fairness. Consequently, models optimized for highly visual or viral content may appear to perform better under this framework. Furthermore, the pipeline is evaluated solely on Twitter/X, without empirical validation across alternative social platforms (e.g., Reddit, Discord, or YouTube), thereby limiting the generalizability of the proposed approach and undermining its claim to universality.

2. Although ECHO is described as “re-runnable,” the paper lacks sufficient methodological transparency for full replication. Critical details concerning data acquisition—such as API endpoints, scraping protocols, temporal sampling strategies, and filtering heuristics—are not disclosed. While Figure D.1 enumerates example keywords, key parameters governing data retrieval, LLM-based relevance scoring, and post-filtering thresholds remain unspecified. This omission impedes reproducibility and may also raise compliance concerns regarding Twitter/X’s data use policies. Moreover, the reuse of user-generated content (including images and textual comments), even in anonymized form, poses potential ethical and legal challenges under data protection regulations such as the General Data Protection Regulation (GDPR).

3. Despite the use of an ensemble of evaluators (GPT-4o, Gemini, and Qwen), the low Kendall’s τ correlation with human judgments (τ ≈ 0.10–0.12) indicates that current VLM-as-a-judge paradigms remain insufficiently reliable for fine-grained assessment of generative quality. Additionally, although approximately 35,000 posts are collected, only ~1,700 high-quality samples are retained for benchmarking. This limited subset, especially when contrasted with larger-scale datasets such as DiffusionDB (≈14 million prompts), raises concerns regarding statistical robustness and the stability of model performance differences. It remains unclear whether the reported distinctions reflect genuine capability gaps or are artifacts of small-sample variance.

**Questions:**

NA

---

> ### Author Response · Authors · 2025-11-26
> **Response to Reviewer Fp5T**
>
> Thank you for your comments! Below, we explore extending ECHO to alternative social platforms, clarify data acquisition details, discuss the safeguards we implemented to address ethical and legal considerations, compare the correlation of other evaluators with human judgements, and validate the stability of model performance differences.
>
> ---
>
> 1. *ECHO’s exclusive reliance on Twitter/X introduces substantial platform- and demographic-specific biases*
>
> **We empirically analyze ECHO’s task distribution and compare it to prior datasets in Sec. A.1 of the Appendix.** Compared with prior image-to-image datasets limited to object and scene editing, ECHO surfaces new tasks like “YouTube Thumbnail Design” and “3D Rendering and Retexturing” (see Fig. A.1). While prior text-to-image datasets focus on generating objects or fantastical imagery, ECHO also tests real-world applications like “Educational Visual Storytelling” and "Food and Product Photography" (see Fig. A.2). Overall, we did not observe any unusual biases in ECHO’s task distribution, compared with prior benchmarks.
>
> ---
>
> 2. *[E]mpirical validation across alternative social platforms*
>
> **We also show that it is straightforward to apply the same ECHO pipeline to Douyin, a non-English and short-form video platform that is very different from Twitter/X.** Many Douyin users post screenshots of their conversations with 4o Image Gen, which the “Conversation Screenshot” step of our pipeline is able to automatically extract.
>
> In a small beta test, we collected 100 Douyin posts and postprocessed them with the ECHO pipeline, with the following quality classifications: {Benchmark: 18, Analysis: 41, Discard: 41}. We also show examples in Fig. A.3 of the Appendix.
>
> ---
>
> 3. *Critical details concerning data acquisition [...] are not disclosed.*
>
> We plan to publicly release the code for the ECHO pipeline for reproducibility, and include an anonymized version here: https://anonymous.4open.science/r/Echo-pipeline-1081. Details on API endpoints, scraping protocols, and key parameters governing data retrieval can be found in `collect_data.py`. Temporal sampling strategies and keywords are specified in `keywords.py`. Post filtering thresholds are specified in `postprocess.py`.
>
> **This code for data acquisition complies with the official Twitter/X API, and follows the same procedures as prior work published at ICLR 2024 [1].**
>
> [1] Srivatsan et. al. Alt-Text with Context: Improving Accessibility for Images on Twitter. ICLR 2024. https://openreview.net/forum?id=97Dl82avFs&noteId=EhjzECJw4c
>
> ---
>
> 4. *[T]he reuse of user-generated content [...] poses potential ethical and legal challenges*
>
> User-generated content indeed poses important ethical and legal considerations. We follow standard protocols to ensure compliance for when our dataset is released, upon publication.
>
> - **First, we will only release post IDs and annotations derived from our pipeline.** The raw images and text from the original post must be re-hydrated using the official Twitter/X API. This is the same release approach as [1], “a new dataset of 371k images paired with alt-text and tweets scraped from Twitter,” which underwent the ethics review process and was subsequently accepted as an ICLR 2024 poster.
> - **Second, we will include a form complying with the right to erasure**, where individuals can request the deletion of samples from the benchmark.
> - **Third, for all samples we apply LLama-Guard-4-12B** [2], a multimodal safety classifier designed according to the MLCommons hazards taxonomy [3], and remove any samples with text or images flagged to contain any of its hazard categories, such as privacy-violating, hateful, or NSFW content.
>
> By implementing these safeguards that are widely accepted for university-led and non-commercial scientific research, we believe we have comprehensively addressed the relevant ethical and legal considerations.
>
> [1] Srivatsan et. al. Alt-Text with Context: Improving Accessibility for Images on Twitter. ICLR 2024. https://openreview.net/forum?id=97Dl82avFs&noteId=EhjzECJw4c \
> [2] Llama Team. Llama guard 4 model card. https://huggingface.co/meta-llama/Llama-Guard-4-12B. \
> [3] Ghosh et. al. Ailuminate: Introducing v1.0 of the ai risk and reliability benchmark from
> Mlcommons. arXiv 2025.

---

> ### Author Response · Authors · 2025-11-26
> **Response to Reviewer Fp5T**
>
> 5. *VLM-as-a-judge paradigms remain insufficiently reliable for fine-grained assessment of generative quality*
>
> We would like to underscore that evaluating the quality of image generation remains a longstanding open problem, and it is not only very difficult but also out of scope for our work. We mainly follow the practice adopted in recent literature (e.g., Bagel), which also uses a VLM-as-a-judge to score generated images.
>
> To address the concern about low VLM-human correlation, we additionally report three widely used automatic metrics: the CLIP text-image similarity score that measures Faithfulness (See Table 1), the FID score that measures Fidelity (See Table 2), and the DreamSim image-image similarity score that measures Diversity (See Table 3). We additionally conduct a human-correlation study under the same setting for the CLIP text-image similarity score (See Table 4) and the DreamSim score (See Table 5).
>
> **As shown, the CLIP score and DreamSim score have similarly weak or even worse correlation with human preferences compared to VLM-as-a-judge. In other words, replacing the VLM judge with CLIP or DreamSim does not solve the problem; all current automatic metrics struggle to directly reflect fine-grained human preference.**
>
> **Table 1. Faithfulness (CLIP Score).** Higher scores means better text-image faithfulness.
> |Model|Text-to-Image|Image-to-Image|
> |-|-|-|
> |4o Image Gen|0.3672|0.2703|
> |Gemini|0.3555|0.2561|
> |Nano Banana|0.3574|0.2461|
> |LLM+Diffusion|0.3468|0.1675|
> |Flux Kontext|0.3287|0.2604|
> |Bagel-Think|0.3281|0.2636|
> |Bagel|0.3218|0.2717|
> |Anole|0.2598|0.2018|
>
> **Table 2. Fidelity (FID Score).** Lower scores mean better visual fidelity. For the reference set, we use 10k images from the COCO 2017 validation set.
> |Model|Text-to-Image|Image-to-Image|
> |-|-|-|
> |4o Image Gen|85.35|83.79|
> |Nano Banana|76.96|70.64|
> |LLM+Diffusion|84.86|77.41|
> |Flux Kontext|87.16|86.04|
> |Bagel|76.21|85.88|
>
> **Table 3. Diversity (DreamSim).** Higher scores mean better diversity.
> |Model|Text-to-Image|Image-to-Image|
> |-|-|-|
> |4o Image Gen|0.1946|0.1139|
> |Nano Banana|0.2912|0.1568|
> |LLM+Diffusion|0.3479|0.3096|
> |Flux Kontext|0.2827|0.1174|
> |Bagel|0.3484|0.1920|
>
> **Table 4. CLIP score:**
> ||τ|p|
> |-|-|-|
> |Image-to-Image|0.0936|0.0089|
> |Text-to-Image|0.0751|0.0221|
>
> **Table 5. DreamSim score:**
> ||τ|p|
> |-|-|-|
> |Image-to-Image|–0.1222|0.986|
> |Text-to-Image|–0.0315|0.746|
>
> ---
>
> 6. *This limited subset, especially when contrasted with larger-scale datasets such as DiffusionDB (≈14 million prompts), raises concerns regarding statistical robustness and the stability of model performance differences.*
>
> Very large-scale datasets, such as DiffusionDB with 14 million prompts, cannot be run in a reasonable amount of time. For example, generating 777 images with Bagel takes ~18 hours on 2 A6000 GPUs. **Most standard image generation benchmarks consist of only a few hundred samples; for example, ImageReward [4] selects a subset of 466 DiffusionDB prompts for testing.** Furthermore, ECHO contributes new prompts significantly different from those in DiffusionDB. These differences can be observed by comparing ECHO and ImageReward (a subset of DiffusionDB) in Fig. 3 of the main paper and Fig. A.2 of the Appendix.
>
> Regarding the stability of model performance differences, we conduct an ablation experiment on the number of output images evaluated, reported in Tables 5 and 6 below. Specifically, we evaluate between 1k and 10k outputs by varying the number of outputs generated per sample, and we found that the results converge to the same value as the number of images increases.
>
> **Table 5. Text-to-Image winrate when varying number of outputs per sample.** Winrates are computed from 1k-10k total images (for O=1 to O=10).
> |Model|O=1|O=2|O=3|O=4|O=5|O=6|O=7|O=8|O=9|O=10|
> |:-|:-:|:-:|:-:|:-:|:-:|:-:|:-:|:-:|:-:|:-:|
> |4o Image Gen|0.70|0.71|0.71|0.72|0.72|0.72|0.72|0.72|0.72|0.72|
> |Nano Banana|0.69|0.70|0.70|0.70|0.70|0.70|0.70|0.70|0.71|0.71|
> |LLM+Diffusion|0.52|0.52|0.52|0.52|0.52|0.52|0.52|0.52|0.52|0.52|
> |Bagel|0.31|0.30|0.29|0.28|0.28|0.28|0.28|0.28|0.28|0.28|
> |Flux Kontext|0.28|0.28|0.28|0.29|0.29|0.29|0.29|0.29|0.29|0.29|
>
> **Table 6. Image-to-Image winrate when varying number of outputs per sample** Winrates are computed from 777-7.7k total images (for O=1 to O=10).
>
> |Model|O=1|O=2|O=3|O=4|O=5|O=6|O=7|O=8|O=9|O=10|
> |:-|:-:|:-:|:-:|:-:|:-:|:-:|:-:|:-:|:-:|:-:|
> |4o Image Gen|0.82|0.81|0.81|0.81|0.81|0.82|0.82|0.81|0.81|0.81|
> |Nano Banana|0.64|0.63|0.62|0.62|0.62|0.62|0.62|0.62|0.62|0.62|
> |Bagel|0.45|0.46|0.46|0.46|0.46|0.46|0.46|0.46|0.46|0.46|
> |Flux Kontext|0.42|0.43|0.43|0.43|0.43|0.43|0.43|0.43|0.43|0.43|
> |LLM+Diffusion|0.17|0.18|0.18|0.18|0.18|0.18|0.18|0.18|0.18|0.18|
>
> [4] Xu et. al. ImageReward: learning and evaluating human preferences for text-to-image generation. NeurIPS 2023.

---

### Official Review · Reviewer_vt4s · 2025-11-11

**Soundness:** 3
**Presentation:** 3
**Contribution:** 3
**Rating:** 6
**Confidence:** 3

**Summary:**

Existing Benchmarks often mirror the capabilities of models to evaluate and understand how models
perform on tasks of interest. These benchmarks contain short and simple instructions. Authors present ,
ECHO framework for constructing benchmarks using 35000 prompts created from real world (social
media data- Twitter) and used GPT4o for image generation to identify SoTA model capabilities and its
alternatives , design metrics , complex task which was absent in existing or previous benchmarks. Curated
prompts were filtered by filtered posts by designing the relevant keywords and relevance of the post text.
Further online discussion may have messy conversation which leads to incorrect dataset collection . To
avoid this authors have used LLMs to turn every messy conversation into one clean, self-contained data
sample that includes: the prompt, the responses, and how good it is. After text refinement , Multimedia
(Images) requires VLM (Qwen 2.5 VL). After Analysis , 20% Data marked as High Quality
(Benchmarking) , 66% Data as Medium Quality (Analysis). Due to policy of twitter and openai some
posts were refused to download or generate and further data removed manually by the author . Following
types of models used such as Unified Models , LLM+Diffusion , Image Editing Models. Instead Human
Evaluation for large scale authors used VLM as a Judge. As a secondary Validation , authors present five
expert raters with outputs of 8 models for 200 samples and ask the annotators to rank the output from best
to worst for both image to image and image to text splits. Authors framed ECHO could help to
differentiate model performance in the fine grained ways such as Color Shift Magnitude , Face Identity
Similarity , Visual structure such as object positioning or human pose.

**Strengths:**

1. Paper primary goal is to create a new generative model structured dataset involving users sharing
interesting prompts and outputs, novel task ideas, or commentary on model behavior.
2. Authors present , ECHO framework for constructing benchmarks using 35000 prompts created
from real world (social media data- Twitter).
3. Designed several specialized automated metrics: color shift magnitude, face identity similarity,
structure distance, and text rendering accuracy.

**Weaknesses:**

1. Community perceptions and discussion topics evolve over time. Benchmarks derived from a
snapshot of community feedback may not adapt quickly, risking obsolescence or misalignment
with current priorities unless actively maintained.

2. Since social media discussions are shaped by active communities, the collected prompts and
feedback may reinforce prevailing stereotypes, biases, or misconceptions. This can lead to
benchmarks that unintentionally favor certain demographic groups, styles, or cultural contexts,
thereby limiting fair assessment across diverse user groups.
3. Leveraging social media data introduces vulnerability to manipulation, such as spam posts,
coordinated misinformation, or artificially amplified feedback, which could skew the benchmark
toward certain failure modes or artificially elevate the perceived performance.

**Questions:**

1. The success of your approach relies heavily on LLMs and CV models for classification and
extraction. How sensitive are these models to inaccuracies or biases, and what measures are in
place to validate their outputs? Could errors in automated classification impact the reliability of
the derived metrics?
2. Are there classes of failures or use cases that ECHO systematically misses due to its reliance on
social media posts or community feedback?
3. Are there risks that users might intentionally or unintentionally manipulate community feedback
(e.g., spamming certain prompts or promoting specific outputs) to bias the benchmark? How does
your framework mitigate or detect such scenarios?

---

> ### Author Response · Authors · 2025-11-26
> **Response to Reviewer vt4s**
>
> Thank you for your insightful points! Below, we explore extending ECHO to non-English platforms with different demographic groups, verify that our benchmark does not exhibit signs of manipulation, and validate the pipeline’s accuracy.
>
> ---
>
> 1. *Community perceptions and discussion topics evolve over time.*
>
> ECHO contributes not only a benchmark but also a re-runnable framework, which addresses this challenge that community perception evolves over time. We plan to publicly release our code such that future work can reuse the ECHO pipeline; an anonymized version is available at: https://anonymous.4open.science/r/Echo-pipeline-1081.
>
> ---
>
> 2. *The collected prompts and feedback may reinforce prevailing stereotypes, biases, or misconceptions*
>
> **To further characterize the prompts within ECHO, we perform a more granular analysis of task clusters in ECHO as well as prior datasets, added to Sec. A.1 of the Appendix.** Compared with prior image-to-image datasets limited to object and scene editing, ECHO surfaces new tasks like “YouTube Thumbnail Design” and “3D Rendering and Retexturing” (see Fig. A.1). While prior text-to-image datasets focus on generating objects or fantastical imagery, ECHO also tests real-world applications like “Educational Visual Storytelling” and "Food and Product Photography" (see Fig. A.2). Overall, we did not observe any unusual biases in ECHO’s task distribution, compared with prior benchmarks.
>
> **Since ECHO is a re-runnable framework, it can also be applied to platforms with other demographic groups and cultural contexts.** We demonstrate this by applying the same ECHO pipeline to Douyin, a non-English and short-form video platform. Many Douyin users post screenshots of their conversations with 4o Image Gen, which the “Conversation Screenshot” step of our pipeline is able to automatically extract. We also show examples in Fig. A.3 of the Appendix.
>
> ---
>
> 3. *Leveraging social media data introduces vulnerability to manipulation*
>
> This is a good point; data poisoning is a widely recognized risk for any benchmark derived from real-world data. In practice, we did not encounter this issue for our proposed benchmark, where we analyze ECHO’s prompts and community feedback in Figures A.5a and A.5b of the Appendix.
>
> To measure possible manipulation, our goal is to detect any unusual peaks or anomalies in the dataset’s text distribution. We first embed the text with OpenAI’s text-embedding-small, compute the centroid as the average of all text embeddings, then compute each embedding’s euclidean distance from the centroid. We visualize this distribution, as well as a Gaussian fit and lines indicating one, two, and three standard deviations from the mean. We observe that both the prompts and community feedback from ECHO exhibit a near-Gaussian distribution with a single peak, with all density within three standard deviations from the mean. **Therefore, Figures A.5a and A.5b demonstrate that ECHO does not exhibit anomalous behaviors like coordinated misinformation or artificially amplified feedback.**
>
> ---
>
> 4. *How sensitive are [LLMs and CV] models to inaccuracies or biases*
>
> We design ECHO such that it is decomposed into clear-cut and modular sub-steps, and empirically validate the automated pipeline’s accuracy in Table 1 below. To quantify accuracy, we take a subset of samples and hand-label the ground-truth for each sub-step. For “Classify Input-Output” we take the unstructured image pool then label each as an input or output, for “Fill-in-the-Blank” we take the template and output image then annotate short keyword completions, for “Quality Filter” we take the sample then label each as Benchmark, Analysis, or Discard (corresponding to high, moderate, and low quality). We also add this analysis to Sec. A.5 of the Appendix.
>
> **As seen in Table 1, each automated sub-step achieves at least 80\% accuracy.** Furthermore, for the final benchmark evaluated in Sec. 5, we perform an additional manual validation step where we discard any incorrectly processed samples. Therefore errors in automated classification do not impact the reliability of the derived metrics.
>
> **Table 1. ECHO pipeline accuracy, computed against 100 human-labeled examples.**
> | | Description | Accuracy |
> |-|-|-|
> | Classify Input-Output | % correct classification in {Input, Output} | 0.9200 |
> | Fill-in-the-Blank | % ground-truth keywords matched | 0.8382 |
> | Quality Filter | % correct classification in {Benchmark, Analysis, Discard} | 0.8000 |
>
> ---

---

> ### Author Response · Authors · 2025-11-26
> **Response to Reviewer vt4s**
>
> 5. *Are there classes of failures or use cases that ECHO systematically misses due to its reliance on social media posts or community feedback?*
>
> You’re right that our reliance on social media data could introduce limited coverage. For example, social media content moderation means that ECHO could systematically miss samples flagged by the platform as violent, hateful, or sensual in nature. Future work could explicitly collect these domains on non-moderated platforms, if they are of interest, while making sure to implement the appropriate safeguards.

---

### Author Response · Authors · 2025-12-01

Dear AC,

Below, we summarize the reviews and rebuttal for your convenience.

Reviewers note that our framework for curating image generation benchmarks using “community-generated evidence from social media as a source” is a “novel solution” (eprr). The resulting benchmark is highlighted as “unique [and] of utmost importance” (dbsG), “highly significant” (eprr), and “timely” (Fp5T, eprr), with a “modular, well-illustrated, and generalizable” pipeline (Fp5T). Our work also contributes the “interesting idea” of “turning user complaints into quantitative metrics” (Fp5T), which “uncover[s] significant model failures” (dbsG).

**Most of the reviews were positive, and Reviewer dbsG suggested a few supplementary experiments for completeness rather than raising concerns about the core contribution. We thoroughly addressed the suggestions in our rebuttal:**
- In Tables 4, 5 we demonstrate that VLM-as-a-judge exhibits a higher human correlation than other widely used metrics like CLIP scores.
- In Tables 6, 7 we clarify the number of images generated per model and ablate this hyperparameter.
- In Tables 8, 9 we report model performance on individual tasks.
- In Tables 1, 2, 3 we report additional Fidelity, Faithfulness, and Diversity metrics.
- In Tables 10, 11 we report additional Counting and Hallucination metrics.

We also plan to publicly release our pipeline, following the same protocols as prior work published at ICLR 2024, with an anonymized version here: https://anonymous.4open.science/r/Echo-pipeline-1081.

Our rebuttal was posted shortly before the OpenReview reviewer/AC leaks, so the reviewers did not have the opportunity to revise their score. We hope our rebuttal addresses all of the reviewer comments, and we appreciate your time and effort in handling our manuscript.

Sincerely,\
Authors

---

### Meta-Review · Area_Chair_UE81 · 2026-01-06

**Summary:**

Reviewers were broadly positive and viewed the paper as novel, timely, and highly significant. The main concerns were requests for additional analyses and clarifications, particularly around evaluation metrics, per-task performance breakdowns, and experimental completeness. Importantly, no reviewer raised concerns about the core idea, methodology, or validity of the proposed benchmark and pipeline. These issues were primarily about strengthening empirical evidence rather than questioning the contribution.

**Reviewer Concerns:**

The rebuttal mostly addressed all substantive reviewer concerns by adding new experiments, metric ablations, per-task evaluations, and clarifications on generation protocols. The justification of VLM-as-a-judge via human correlation analysis was particularly convincing. The planned public release of the pipeline further strengthens the paper’s impact. No major concerns remain outstanding; any remaining limitations are minor and do not affect the acceptance decision.

**Reviewer Scores:**

The initial scores of 3 out of 4 reviewers were 6 (marginally above the acceptance threshold). Post rebuttal the reviewer who gave 4 would have increased the score, indicating that all reviewers would have stayed at 6 score post rebuttal.

---

### Decision · Program_Chairs · 2026-01-26

Accept (Poster)